# Comparisons between the distributions of dust and combustion aerosols in MERRA-2, FLEXPART, and CALIPSO and implications for deposition freezing over wintertime Siberia

Lauren M. Zamora[1,2], Ralph A. Kahn[2], Nikolaos Evangeliou[3], Christine D. Groot Zwaaftink[3], and Klaus B. Huebert[4]

[1]Earth System Science Interdisciplinary Center, University of Maryland, College Park, College Park, Maryland, U.S.A.
[2]NASA Goddard Space Flight Center, Greenbelt, Maryland, U.S.A.
[3]Norwegian Institute for Air Research (NILU), Kjeller, Norway
[4]CSS, Inc., Fairfax, Virginia, U.S.A.

*Correspondence to*: Lauren M. Zamora (lauren.m.zamora@nasa.gov)

**Abstract.** Aerosol distributions have a potentially large influence on climate-relevant cloud properties but can be difficult to observe over the Arctic given pervasive cloudiness, long polar nights, data paucity over remote regions, and periodic diamond dust events that satellites can misclassify as aerosol. We compared Arctic 2008-2015 mineral dust and combustion aerosol distributions from the Cloud-Aerosol Lidar and Infrared Pathfinder Satellite Observation (CALIPSO) satellite, the Modern-Era Retrospective analysis for Research and Applications, Version 2 (MERRA-2) reanalysis products, and the FLEX-ible PARTicle (FLEXPART) dispersion model. Based on coincident, seasonal Atmospheric Infrared Sounder (AIRS) Arctic satellite meteorological data, diamond dust may occur up to 60% of the time in winter, but it hardly ever occurs in summer. In its absence, MERRA-2 and FLEXPART each predict the vertical and horizontal distribution of large-scale patterns in combustion aerosols with relatively high confidence (Kendall Tau rank correlation > 0.6), although a sizeable amount of variability is still unaccounted for. They do the same for dust, except in conditions conducive to diamond dust formation where CALIPSO is likely misclassifying diamond dust as mineral dust, and near the surface (< ~2 km) where FLEXPART may be overpredicting local dust emissions. Comparisons to ground data suggest that MERRA-2 Arctic dust concentrations can be improved by the addition of local dust sources. All three products predicted that wintertime dust and combustion aerosols occur most frequently over the same Siberian regions where diamond dust is most common in the winter. This suggests that dust aerosol impacts on ice phase processes may be particularly high over Siberia, although further wintertime model validation with non-CALIPSO observations is needed. This assessment paves the way for applying the model-based aerosol simulations to a range of regional-scale Arctic aerosol-cloud interaction studies.

# 1 Introduction

Aerosols have a potentially large influence on Arctic climate-relevant cloud properties such as cloud cover, cloud phase, and cloud particle size (Alterskjær et al., 2010; Gagné et al., 2017; Morrison et al., 2012; Schmale et al., 2021; Shindell and Faluvegi, 2009; Willis et al., 2018; Zamora et al., 2018). Mineral dust, for example, is thought to be a particularly efficient ice nucleating particle (INP) source, leading to enhanced freezing of liquid aerosol particles or potentially to depositional growth under favorable environmental conditions (Kanji et al., 2017). Marine aerosols are a source of both cloud condensation nuclei

(CCN) and INPs (Willis et al., 2018). Combustion (anthropogenic pollution + biomass burning) aerosols are less efficient INPs (Kanji et al., 2017), but can readily form CCN, as can sulphate aerosols.

However, understanding the effects of aerosols on Arctic clouds is limited in large part by uncertainties in the distributions of aerosols and the contributions of each aerosol type to the CCN and INP budgets. Ground-based aerosol observations are sparse, and aerosol data from passive satellite instruments are unavailable during polar night (Duncan et al., 2020). Long-term,

vertically resolved remote sensing information on Arctic aerosols, including aerosol subtype distributions from dust, smoke, and other sources are available from the Cloud-Aerosol Lidar and Infrared Pathfinder Satellite Observation (CALIPSO) satellite (Di Pierro et al., 2013; Winker et al., 2013; Kim et al., 2018; Yang et al., 2021; Chen et al., 2021). However, CALIPSO can miss aerosols that are a) dilute (Kacenelenbogen et al., 2014; Zamora et al., 2017; Di Pierro et al., 2013; Winker et al., 2013; Rogers et al., 2014), b) very small (Hallen and Philbrick, 2018), including highly numerous marine biogenic aerosols

(Burkart et al., 2017), c) in the 200 m immediately above the surface where local marine and terrestrial emission concentrations are highest (Winker et al., 2013), and d) below clouds. This latter issue is particularly challenging, as clouds occur in the Arctic at least half the time during winter, and up to 80% of the time over open ocean in summer (Zygmuntowska et al., 2012). Moreover, CALIPSO does not measure lidar ratios, and so periodic Arctic diamond dust events (i.e., ice crystal precipitation in clear sky conditions) can sometimes be misclassified as mineral dust aerosol, leading to overestimates of dust aerosol

presence (Di Biagio et al., 2018).

Models are therefore critical for providing aerosol estimates, especially below and within clouds at all vertical levels. However, most aerosol models are poorly validated at high altitudes and latitudes (Arnold et al., 2016; Eckhardt et al., 2015). These combined remote sensing and model issues lead to large uncertainties in predictions of Arctic aerosol types, levels, and their resulting cloud impacts now and in the future.

In this paper, we aim to take advantage of the complementary information from model and reanalysis products and satellite data to a) understand the strengths and limitations of the model and reanalysis products and b) better identify those Arctic regions with the highest certainty and uncertainty in the distributions of dust and combustion aerosols. This information will enable improvements in the model and reanalysis products and will facilitate a range of Arctic aerosol-cloud interaction studies going forward, including targeted suborbital measurements. As one application of these products, we use the overlap between

meteorological conditions conducive to deposition nucleation and elevated aerosol presence in CALIPSO, the Modern-Era

Retrospective analysis for Research and Applications, Version 2 (MERRA-2), and the FLEXPART (FLEX-ible PARTicle) Lagrangian particle dispersion model to show that this process may be more common during winter than previously thought.

## 2 Methods

This study focuses on the Arctic areas poleward of 60 °N between 2008 to 2015. In some cases, data were separated into terrestrial and oceanic regions, as defined by the ETOPO1 1 Arc-Minute Global Relief Model (Amante and Eakins, 2009)). We assess the commonalities and differences in dust and combustion aerosol distributions between CALIPSO, MERRA-2 reanalysis aerosol products, and aerosols in the FLEXPART model. All data analysis was performed using the R language and environment for statistical computing (R Core Team, 2022).

### 2.1 Data sources

#### 2.1.1 CALIPSO data

Aerosol data, including aerosol layer base and top elevation, cloud-aerosol discrimination (CAD) score, and aerosol type, were obtained from the CALIPSO Lidar Level 2 5-km Merged Layer Data, V4-20 dataset (Winker, 2018), as were data on solar zenith angle (SZA). CALIPSO reports an aerosol vertical resolution of 30 m up to ~8.2 km above sea level (ASL), and 60 m above that up to 20.2 km ASL. Averaged aerosol horizontal resolution ranges between 5 and 80 km, with higher resolution at higher aerosol concentrations, when signal-to-noise ratios are better (Vaughan et al., 2009). We focused on data above 200 m from the surface, to reduce the influence of blowing snow and ground contamination of the CALIPSO lidar (Winker et al., 2013) on the results.

For this analysis, CALIPSO aerosol data were used only from cloud-free profiles where CAD scores ranged between -100 and -30, to exclude clouds and very low-confidence aerosol layers. CALIPSO aerosol layer properties have higher uncertainties during the daytime, especially over bright sea ice. As mentioned previously, CALIPSO may also miss dilute Arctic aerosols, even at night, when lidar sensitivity is higher (Zamora et al., 2017). Moreover, CALIPSO can be subject to other errors in aerosol subtype designation (Omar et al., 2009; Kim et al., 2018). In general, aerosol type is more difficult to discern in aerosol mixtures (Zeng et al., 2021). CALIPSO does not consider a marine aerosol category in over-land retrievals, but these aerosols may still be present, at least near coastal locations (Kanitz et al., 2014). Also, desiccated marine aerosols might be misclassified as dust or polluted aerosol at relative humidity below 60-70% (Ferrare et al., 2020), and pollution aerosols can be misclassified as marine aerosols (Di Biagio et al., 2018). Furthermore, there is a risk that some layer classification results can be confounded by vertical averaging over layers of different aerosol types, such as dust lying over marine aerosol layers.

Evaluations of the previous CALIPSO aerosol type version 3 dataset mainly from subarctic data indicate agreement with aerosol type estimates from other data sources most of the time, with best results for mineral dust aerosols (Burton et al., 2013; Papagiannopoulos et al., 2016; Mielonen et al., 2013). However, in the Arctic, diamond dust can be misattributed to the

CALIPSO dust aerosol subtype (Di Biagio et al., 2018), and so CALIPSO dust subtyping is likely less certain over this region. It also appears that polluted continental aerosols over the Arctic Ocean may be misattributed to clean marine conditions (Rogers et al., 2014; Di Biagio et al., 2018), although this may not matter at all locations, as the seasonality of the clean marine aerosol

subtype seems to be in agreement with long-term observations at Svalbard (Di Biagio et al., 2018). In this study, we used the updated version 4 products for aerosol type classification (Kim et al., 2018), which may result in fewer uncertainties in CALIPSO aerosol type compared to previous studies. However, large-scale evaluations of CALIPSO version 4 aerosol subtype have not yet been conducted to our knowledge, and even for version 3, such evaluations are limited.

Any CALIPSO aerosol layer in cloud-free conditions that was classified as either "dust," "polluted dust," or "dusty

marine" was included in a larger dust group for comparison to other datasets. Any CALIPSO aerosol layer classified as either "polluted continental/smoke", "polluted dust," or "elevated smoke" was included in a larger combustion aerosol group. "Polluted dust" was included in both the larger dust and pollution groupings, reflecting the fact that aerosols of both types can be mixed in the atmosphere. Polluted dust made up 45% of the dust cases, and 39% of the combustion cases, and so there is significant overlap between the two groups. We discuss how removing the "polluted dust" component of the combustion

aerosol group effects the results in section 3.3.

### 2.1.2 AIRS data

CALIPSO profiles were matched with concurrent temperature (T) and relative humidity (RH) data from the Atmospheric Infrared Sounder (AIRS) L3 Daily Standard Physical Retrieval (AIRS+AMSU) 1˚ x 1˚ V007 (AIRX3STD 007) product (AIRS project, 2019) from both the ascending and descending orbits. These data are available in 12-hourly time slots

for every 1˚ x 1˚ at pressure levels 1000, 925, 850, 700, 600, 500, 400, 300, 250, 200, 150, and 100 mb. The AIRS L3 data are useful under most conditions in the Arctic troposphere but have fewer errors when there is no heavy precipitation.  Data with RH values > 200% were discarded, following AIRS team recommendations (Tian et al., 2020). Seasonally averaged AIRS data during the study period were obtained separately using the Giovanni online data system (NASA GES DISC, Acker and Leptoukh (2007). For the seasonal data we used the ascending and descending orbits in the AIRX3STM 007

product.

Ice crystals, including diamond dust (Table 1), form and grow in the atmosphere differently depending on ambient ice nuclei, temperature, and moisture levels. To identify locations where diamond dust is most likely to form in the presence of aerosols, we follow a similar approach as in Sakai et al. (2003), based on the locations where the equilibrium relative humidity over ice is exceeded (i.e., where ambient values of RH with respect to ice ($RH_i$) are supersaturated, or > 100%). $RH_i$

values were calculated from AIRS T and RH values following Murphy and Koop (2005). First, saturation vapor pressure over liquid water ($e_s$) and saturation vapor pressure with respect to ice ($e_{si}$) were calculated from the AIRS T values based on the Murphy and Koop equations 10 and 7, respectively. These equations are valid for 123 < T < 332 K and down to 110 K, respectively. Then we estimated $RH_i$ by multiplying this ratio by the relative humidity, following their equation 11:

$$RH_i = RH \frac{e_s}{e_{si}} \qquad (1)$$

This approach could underestimate locations where diamond dust occurs, as it does not include, for example, locations with homogeneous freezing of preactivated aerosol pore water at temperatures < -38 °C (Marcolli, 2014) (Table 1). Locations where diamond dust forms from small-scale meteorological variations in supersaturation from factors such as vertical velocity (Korolev and Mazin, 2003) and radiative cooling (Zeng, 2018) will also be underestimated with this method.

**2.1.3 MERRA-2 output**

       Mineral dust, black carbon (BC), and organic carbon (OC) aerosol concentrations and model mid-layer height output were obtained from MERRA-2 (Global Modeling and Assimilation Office (GMAO), 2015a, b), which has 3-hourly, 0.5 ° x 0.625 ° horizontal resolution. We obtained output at 72 different model levels above the surface, focusing mainly on the lower 29 levels (up to ~10.5 km ASL). MERRA-2 aerosol emissions datasets are described in Table 1 of Randles et al. (2017), and

do not include local Arctic mineral dust sources. Aerosols are assumed to be externally mixed with different components (e.g., BC, OC, sulfate aerosols) each contributing to total aerosol load. Aerosol loss processes include dry and wet deposition (Randles et al., 2017), with precipitation-induced aerosol deposition based on merged precipitation observations and model products in MERRA-2 (Reichle et al., 2017). MERRA-2 assimilates aerosol data when available. During the time period of this study, MERRA-2 assimilated aerosol information from the Aerosol Robotic Network (AERONET), Multi-angle Imaging

SpectroRadiometer (MISR), and Moderate Resolution Imaging Spectroradiometer (MODIS) instruments (Randles et al., 2017). Data from these instruments were assimilated throughout the year in the subarctic, and a fraction of these aerosols was then transported into the Arctic. During the daytime, MERRA-2 also assimilates some limited Arctic aerosol data in non-cloudy conditions. However, those Arctic data sources are unavailable during polar night. As a result, MERRA-2 aerosol output is more model-driven during polar night. Uncertainties may be largest near the surface, which is more impacted by local

sources, than in the middle and upper troposphere, which is more influenced by transported aerosols.

       Mineral dust mass is modelled in five particle size bins with diameters between 0.2-2, 2-3.6, 3.6-6, 6-12, and 12- 20 µm (Colarco et al., 2010). Dust emissions are wind-driven for each size bin (Randles et al., 2017), parameterized following Marticorena and Bergametti (1995). We used dust from the five size classes and grouped them together for comparison with FLEXPART dust aerosols. Note that as with many other dust models, particles with diameters >20 µm are not assessed. Such

larger dust particles are comparatively rare but are observed in the field (Weinzierl et al., 2017; Drakaki et al., 2022) and so their exclusion could lead to an underestimate of *in situ* dust concentrations, especially near local sources.

       BC and OC are represented with two (hydrophobic and hydrophilic) mass tracers, which we added together. BC and OC emissions are based on biomass burning and anthropogenic emissions, including from ships (Randles et al., 2017). For reference, the single scattering albedos of BC and OC in MERRA-2 depend on RH, but the range for BC is around 0.3-0.4 at

~500nm (A. Darmenov, pers. comm.). These simulations did not include brown carbon (BrC), which is expected to more or less track the modelled OC from fires. Aerosol sulfate ($SO_4^{2-}$) data were available in MERRA-2 and contribute to combustion plumes from fires and Arctic haze. For example, within MERRA-2, the total aerosol optical depth (AOD) of a fresh smoke plume might roughly be driven by roughly ~90% OC, with the remaining ~10% partitioned between BC and $SO_4^{2-}$, with the portions changing over time as the plume ages. However, $SO_4^{2-}$ aerosol data were not used in this study as additional tracers

for combustion aerosols because there are $SO_4^{2-}$ contributions from other sources such as marine and volcanic emissions.

There are several studies that have evaluated MERRA-2 Arctic aerosol distributions, mainly using MODIS aerosol optical depth and ground-based observations (e.g., Wu et al. (2020), Lee et al. (2020) and Sitnov et al. (2020)). MERRA-2 BC and OC aerosols tend to be a bit high compared to observed aerosol concentrations, although they tend to follow the qualitative trends (Vinogradova et al., 2020; Zhuravleva et al., 2020; Xian et al., 2022). One study found that dust optical depth and dust

extinction were similar or a bit elevated compared to that of CALIPSO, but with large discrepancies in absolute concentrations (both over- and underpredicting concentrations) compared to two ground sites (Wu et al., 2020).

### 2.1.4 FLEXPART output

Separate simulations of mineral dust, BC, and OC were conducted using the FLEXPART version 10.4 Lagrangian particle dispersion model (Pisso et al., 2019). In the simulations presented here, the model was forced by ERA-Interim meteorological

fields from the European Centre for Medium-Range Weather Forecasts (ECMWF) at 1° x 1° spatial and 3-hourly temporal resolution. In addition to dry and wet deposition, FLEXPART accounts for turbulence (Cassiani et al., 2014), unresolved mesoscale motions (Stohl et al., 2005) and includes a deep convection scheme (Forster et al., 2007). Gravitational settling, dry deposition and in-cloud and below-cloud scavenging are also included (Grythe et al., 2017). The resulting daily output has 1° x 1° horizontal resolution with upper vertical layer boundaries at 10, 100, 250, 500, 750, 1000, 1500, 2000, 4000, 6000, 8000,

10,000, 15,000, and 20,000 m above ground level (AGL). For comparison with the other datasets, we converted the FLEXPART output to km ASL using surface elevation data from ETOPO1 bedrock GMT4 data (Amante and Eakins, 2009).

Emissions of mineral dust include local Arctic sources and were calculated with the FLEXDUST emission model (Groot Zwaaftink et al., 2016). Dust aerosols were split in 10 size classes in FLEXPART: 0.2, 0.5, 1, 1.5, 2.5, 5, 7.5, 12.5, 15, 20 μm diameter. Emitted dust is assumed to follow the Kok (2011) size distribution. Dust from the different size classes were then

grouped together for further analysis and comparison to MERRA-2 dust aerosols.

As with MERRA-2, brown carbon was not modelled explicitly, and OC and BC are used as the main proxies for combustion aerosols. BC and OC were run separately, and do not chemically age over time or interact in the model. They were also assumed to be hydrophilic. BC and OC concentrations were calculated from both anthropogenic emissions (using ECLIPSEv6b) and biomass burning (GFED4.1s (Giglio et al., 2013)), following Klimont et al. (2017) but with updated

emissions factors (Z. Klimont, pers. comm.). The tracking of BC and OC particles includes gravitational settling for all spherical particles, and BC and OC aerosols have assumed mean diameters of 0.25 μm, a logarithmic standard deviation of

0.3, and a particle density of 1500 kg m$^{-3}$ (Long et al., 2013). The BC and OC emissions datasets may not include some local sources of combustion aerosols.

Details on FLEXPART Arctic aerosol distributions have been discussed previously (Groot Zwaaftink et al., 2016; Stohl,
2006; Eckhardt et al., 2015) and are further evaluated in section 3 below. For now, we just note that smoke and pollution transport in FLEXPART have been well validated over the Arctic, and various observations suggest that FLEXPART BC can be a proxy for strong, CALIPSO-detectable aerosol layers (Damoah et al., 2004; Eckhardt et al., 2015; Forster et al., 2001; Paris et al., 2009; Sodemann et al., 2011; Stohl et al., 2002, 2003, 2015; Zamora et al., 2017, 2018). In contrast, mineral dust aerosol validation data in the high Arctic are rare, and prior analysis suggests somewhat higher uncertainty in FLEXPART
dust aerosols (Groot Zwaaftink et al., 2017, 2016).

## 2.2 Comparisons between MERRA-2, FLEXPART, and CALIPSO

We next compared dust and combustion aerosol products in MERRA-2, FLEXPART, and CALIPSO. Marine aerosols are
another important aerosol source over the Arctic (Schmale et al., 2021). They are not included in this assessment because of uncertainties in detecting near-surface aerosols and small biogenic marine aerosols, as discussed in section 2.1.1.

It can be challenging to compare modelled aerosol concentrations to CALIPSO aerosol property information on a case-by-case basis. Doing so requires speculative assumptions about the lidar ratio and the extinction cross section of particles that are beyond the scope of this paper. Also, the amount of aerosol needed for CALIPSO to detect an aerosol layer is unknown
and may vary over time and space, and the same can be said for the thickness of a CALIPSO layer required to be comparable with model aerosol concentrations. Therefore, our approach is instead to 1) assess how well large-scale Arctic CALIPSO combustion and dust aerosols are related to those in MERRA-2/FLEXPART, and 2) find and discuss where the largest discrepancies and similarities are between the products.

To begin, we assessed the correlation between MERRA-2/FLEXPART aerosol concentrations and mean CALIPSO
aerosol layer fraction. Correlations were calculated only on cloud-free days. CALIPSO aerosol layer fraction was defined as the fraction of the CALIPSO aerosol layer of the subtype of interest - dust or combustion – within each altitude bin for an individual CALIPSO observation. Then, as CALIPSO has much finer vertical resolution than either MERRA-2 or FLEXPART, CALIPSO dust layer fraction and MERRA-2/FLEXPART dust concentrations were averaged within 20° longitude × 6° latitude bins and at model vertical resolution for FLEXPART, and at 1 km vertical resolution for MERRA-2.
Similarly, CALIPSO combustion aerosol fraction was compared with MERRA-2/FLEXPART BC and OC concentrations. We excluded aerosol layers within 200 m of the surface in the lowest altitude bin and longitude-latitude-altitude bins with < 30 observations (collectively, < 1% of total data). Data were averaged either across the 8-year study period for one season (e.g., December through February) or for daytime/nighttime samples, as stated in the text.

The robust Kendall Tau rank correlation metric ($\tau$) was chosen to assess correlations. It is calculated by examining all
possible combinations of two data points in the data set and scoring them as either concordant (positive slope) or discordant
(negative slope). $\tau$ is defined as:

$$\tau = \frac{C-D}{C+D} \qquad (2)$$

where C and D are the number of concordant and discordant pairs. In the case of ties, where a pair of data points is neither
concordant nor discordant, a small correction is made ($\tau_b$, following Kendall (1945)).

The Kendall Tau correlation metric is a nonparametric, rank-based alternative to $R^2$ that is robust to outliers, makes
no assumptions about the data distributions, and is thus a better metric of correlation for many types of data (Shevlyakov &
Oja 2016). Kendall Tau ranges from -1 for a perfect negative correlation to 0 for no correlation to +1 for a perfect positive
correlation. In cases where the use of $R^2$ would be appropriate, the magnitude of Kendall Tau is a good estimator for $R^2$, with
a maximum theoretical asymptotic difference < $\pm 0.11$ (Shevlyakov and Oja 2016). To weight longitude-latitude-altitude bins
proportionally to their surface areas, the Kendall Tau metric was estimated using a resampling/bootstrapping method, drawing
100,000 values with replacement from the original data set.

        Sometimes CALIPSO had multiple overlapping layers of aerosol subtypes of interest, which is a result of CALIPSO's
multiscale averaging approach (Thorsen et al., 2011). In those cases, we summed of only the portions of each CALIPSO layer
that did not overlap with the other CALIPSO aerosol layers and that fell completely within the MERRA-2/FLEXPART bin.
The sum of these portions was then divided by the entire height of the MERRA-2/FLEXPART bin to provide the fraction of
the model bin filled by a CALIPSO aerosol layer. To provide more trustworthy comparisons, we also focused primarily on
environmental conditions where the CALIPSO satellite product does best. These conditions include cloud-free, nighttime cases
when diamond dust does not occur, at altitudes > 200 m over the surface. However, for comparison, correlations were also
analysed separately during daytime (defined as when the solar zenith angle (SZA) is < 90º) when the lidar signal-to-noise ratio
is smaller, and when RHi is > 100%, when potential aerosol type errors from diamond dust might be highest.

        Next, we wanted to better understand where aerosol distributions between MERRA-2/FLEXPART and CALIPSO have
the highest agreement. For this step, we assessed the difference in Z-scores between MERRA-2/FLEXPART concentrations
and aerosol layer presence in CALIPSO. Z-scores in MERRA-2/FLEXPART for dust, BC, and OC are defined as the number
of standard deviations from the mean respective dust, BC, or OC value across the study region at a given altitude level.
Locations with high aerosol levels in MERRA-2/FLEXPART will have high positive Z-scores, and locations with low aerosols
will have negative Z-scores. Similarly, Z-scores in CALIPSO are defined as the number of standard deviations from the mean
dust or combustion aerosol layer fraction across the study region at a given altitude. Because Z-scores are unitless, they can
be compared between MERRA-2/FLEXPART and CALIPSO, which has different units. This approach also enables relative
comparisons between MERRA-2 and FLEXPART aerosol distribution patterns even if the concentrations are on different
scales.

We focused on 5° longitude × 2° latitude bins at 0.2 to 2 km, 2-4 km, and 4-8 km vertical resolution during the winter season (December to February). Within those bins, we averaged MERRA-2/FLEXPART dust concentrations and CALIPSO dust aerosol presence across the study period and assessed the difference in Z-scores between the two products. Similarly, we compared differences in Z-score between MERRA-2/FLEXPART BC and OC and CALIPSO combustion aerosol layer fraction. The Z-score differences help locate where the overall patterns agree best between the two products. For example, if at a given location, MERRA-2 aerosols were three standard deviations above the mean and CALIPSO aerosol layer presence was only one standard deviation above its mean, that would result in a Z-score difference of two. The closer the Z-score difference is to zero, the more agreement there is between the products.

One benefit of this approach is that we can compare similar numbers of aerosol events with CALIPSO for both models without assuming a priori which of the two models has more accurate absolute aerosol concentrations. This is helpful, given that there is a paucity of in situ validation data over the Arctic at higher latitudes and altitudes, and so it is possible that either MERRA-2 or FLEXPART aerosol concentrations could be biased high or low. Another benefit is that it offers a way to compare aerosol presence between MERRA-2/FLEXPART and CALIPSO, which could not be directly due to their different units.

## 3 Results and Discussion

### 3.1 Lower troposphere wintertime aerosol distributions are elevated over Siberia

Figure 1 shows the average wintertime distribution of MERRA-2 and FLEXPART submicron dust, BC, and OC aerosols at different altitude levels below 4 km at the same locations where CALIPSO cloud-free profiles were available. MERRA-2 predicts higher mineral dust concentrations than FLEXPART, despite not including local dust sources (although wintertime local dust emissions are small when the soil is frozen or covered in snow). MERRA-2 also had slightly higher OC concentrations over North America but FLEXPART BC levels were elevated at most locations relative to MERRA-2. These differences in dust and combustion aerosols may be influenced by different assumptions in emitted particle size distribution and subsequent transport and deposition.

Wintertime CALIPSO aerosol presence in cloud-free profiles at the same altitude levels as the FLEXPART and MERRA-2 results is also shown in Figure 1. CALIPSO aerosol presence is not directly comparable to average MERRA-2 and FLEXPART aerosol concentrations, which for example, can be skewed by high concentrations during infrequent events. However, the general regional trends still provide information on where dust and combustion aerosols are most common. For example, based on Figure 1, CALIPSO dust aerosols are commonly observed below 4 km over Siberia during wintertime. High average dust concentrations are also predicted by MERRA-2 and FLEXPART in this region, providing confidence in elevated dust levels in the region. In contrast, there is some indication from CALIPSO and FLEXPART that dust sources over the western hemisphere may also be slightly elevated, but below 2 km, there was disagreement on whether these sources are

more elevated over the Canadian archipelago (FLEXPART) or over the Labrador Sea (CALIPSO). As there are no known major dust sources over this region, there are higher uncertainties in dust distributions over North America.

CALIPSO, MERRA-2 and FLEXPART also agree that wintertime combustion aerosols are elevated over the European and Asian portions of the Arctic, in line with other studies (e.g., Eckhardt et al. (2015); Di Pierro et al. (2013)), and consistent with smoke sources from these locations. However, combustion aerosol layer distributions below 4 km are more sharply reduced over oceanic areas in CALIPSO than is predicted in the models. This observation is likely caused by a known CALIPSO aerosol subtyping artifact (Burton et al., 2013; Kanitz et al., 2014; Campbell et al., 2013; Papagiannopoulos et al., 2016; Zeng et al., 2021), as a) there are no known dramatic precipitation differences over the land vs. ocean over this large area to explain this phenomenon, b) aerosols are treated differently in the CALIPSO aerosol subtyping algorithm if they are taken over land and ocean (Kim et al., 2018), and c) others have found that polluted continental aerosols over the Arctic Ocean can be misattributed to clean marine conditions (Rogers et al., 2014; Di Biagio et al., 2018). The dust and combustion aerosol trends seen above 4 km are more difficult to discern (Fig. S1), given the inability of CALIPSO to detect very dilute aerosols.

### 3.2 There is overlap between the distribution of aerosols and the meteorological conditions conducive to diamond dust formation

We next assessed the locations where diamond dust occurs, as these are known to lead to overestimation of dust in CALIPSO (Di Biagio et al., 2018). Table 1 shows the pathways through which diamond dust is most likely to form. Based on Table 1, the homogeneous and deposition ice nucleation pathways for diamond dust formation are most likely to occur when $RH_i$ is > 100%. In line with previous studies (Intrieri and Shupe, 2004; Maxwell, 1982), we do not expect diamond dust formed from the Table 1 pathways to be important during the summer (Figure 2b,d). There are some minor differences in $RH_i$ above 4 km during the summer between ocean and land (Fig. 2b), which are likely due to stronger vertical mixing between more moist surface air and colder stratospheric air over land versus ocean in the summer compared to winter (Stohl, 2006), but generally $RH_i$ conditions are well below 100% during summer at most locations and altitudes.

However, conditions favourable for diamond dust formation from these pathways do occur frequently in some locations during the winter, although diamond dust may not always be present and detectable by CALIPSO when conditions are favourable for its formation. At 925 mb, we estimate that conditions favourable to diamond dust formation occur up to 60% of the time (Fig. 2a), especially in low-lying areas of the Canadian Archipelago and the Siberian interior and coast (Fig. 2c). These are extremely cold Arctic locations that also routinely experience moisture transport events (Dufour et al., 2016; Graham et al., 2017), and which are undersampled by ground-based observations in winter when diamond dust is most likely to occur. Diamond dust formation may also occur in the frigid conditions near the wintertime tropopause (Fig. 2b), and is particularly likely to occur very near the surface within the stable boundary layer (Intrieri and Shupe, 2004). This means that wintertime diamond dust is most likely to confound the dust-CALIPSO comparisons at these times and places (as is discussed further in section 3.3 and Figure S2 of the Supplement).

Interestingly, around 1-1.5 km ASL over Siberia there is overlap between where both MERRA-2, and FLEXPART predict
higher mineral dust and combustion aerosols and where high $RH_i$ values also appear. We have some confidence that the dust levels really are elevated in this region because although CALIPSO may misattribute diamond dust to mineral dust at times, MERRA-2 and FLEXPART do not have this source of error. Therefore, based on the data shown in Figures 1 and 2c, we conclude that Arctic dust aerosol impacts on ice and mixed phase processes are particularly high over Siberia during winter.

Conditions are also favourable for diamond dust formation over the Canadian archipelago during winter. The presence of
320 mineral dust in this region in FLEXPART (which includes local dust sources) but not in MERRA-2 (which does not include local dust sources) combined with the uncertainties in dust presence from CALIPSO under conditions favoring diamond dust formation make it harder to determine whether mineral dust aerosols could have a disproportionate impact on ice and mixed phase processes over this region, as is inferred over Siberia.

The locations where we estimate diamond dust formation occurs are consistent with ship-based observations. During the
325 Surface Heat Budget of Arctic Ocean (SHEBA) ship campaign (1997-1998 in the Beaufort Sea), diamond dust was observed 23% of the time between December and February, mainly near the ocean surface (Intrieri and Shupe, 2004). The first wintertime AIRS data over that region were taken starting December 2002, and so are not directly comparable with the SHEBA data. However, based on AIRS data taken during the 2002-2003 winter at 1000 mb within a similar area (between 74.5 and 80.5 °N and -142 and -168 °E, Fig. 2), $RH_i$ exceeded 100% about 26% of the time, which is comparable with how often
diamond dust was observed during SHEBA five years prior. Conditions conducive to diamond dust formation are much less likely to occur in the mid-troposphere in this region (e.g., see distributions in Fig. 2), also in line with SHEBA observations of diamond dust forming predominantly near the surface.

It is important to note that the impacts on clouds of the dust and combustion aerosols of focus in this study may be much smaller than those of local marine emissions, particularly at times when the surface is not covered with snow or ice. For
example, one recent satellite study found evidence that mixed phase cloud formation occurs above homogeneous freezing on average but at colder wintertime temperatures over both the Siberian and Canadian archipelago regions than over other times and Arctic locations (Carlsen and David, 2022), which they attribute to fewer marine ice nucleating particles being emitted there.

Also, the locations where AIRS $RH_i$ values are > 100% may not capture every instance of diamond dust formation.
Diamond dust could occur in locations that are estimated to be subsaturated with respect to ice if, for example, ice particles fall through subsaturated layers, or if the AIRS resolution (reported at 1˚ in the current study) is too coarse to observe smaller-scale supersaturations (Sakai et al., 2003). As previously mentioned, diamond dust may also occur in conditions favourable for freezing of preactivated water-containing aerosol pores at $RH_i$ values < 100% (third line mechanism, Table 1). Such conditions are presumed less likely to occur during the summer due to the warmer average temperatures, but their impact
during colder periods of the year is currently not well known.

## 3.3 Model aerosol evaluation

Next, we assess how well large-scale aerosol distributions in MERRA-2 and FLEXPART compare to those in CALIPSO. Figure 3 shows the MERRA-2 and FLEXPART Arctic area-averaged Kendall Tau correlation with mean CALIPSO aerosol layer fraction during night and day (all data throughout the year where SZA is either > or < 90°, respectively). For reference, Figure S3 shows corresponding $R^2$ values. We primarily focus the subsequent discussion on times and locations when CALIPSO data are most trustworthy, i.e., during nighttime when $RH_i$ < 100% with "polluted dust" contributing to the overall dust and combustion aerosol groups in CALIPSO (grey background, solid lines, left two columns in Figure 3). It is important to note that even for these carefully selected conditions, limitations in the number of aerosol species being modelled vs. observed can contribute to unexplained differences between MERRA-2/FLEXPART and CALIPSO. Real world CALIPSO aerosol layer mixtures contain additional species that may vary relative to the modeled constituents. For example, non-carbonaceous constituents in a combustion plume observed by CALIPSO can be present at high or low ratios relative to the modelled BC and OC and be mis-classified as carbonaceous. As another example, carbonaceous, biogenic, sulphate or maritime aerosols can be present in a dust plume. These constituents might lead to a bias in the MERRA-2/FLEXPART/CALIPSO comparisons at certain locations (e.g., over the open ocean surface downwind of a continental dust source).

Comparisons are shown for other conditions as well. For example, from Figure 3, it is immediately apparent that the relationship between MERRA-2/FLEXPART and CALIPSO dust diminishes substantially in the presence of diamond dust, likely due to CALIPSO misclassification of diamond dust as mineral dust. This hypothesis is supported by the fact that CALIPSO observed Arctic dust layers on average $61 \pm 11\%$ more frequently in wintertime air masses in conditions favourable for diamond dust formation (RHi > 100%) (Fig. S2). A similar trend is also observed to a smaller extent for combustion aerosols (Fig. 3).

However, outside of diamond dust conditions, nighttime MERRA-2 and FLEXPART aerosol concentrations and CALIPSO aerosol layer presence are moderately well correlated, based on the Kendall Tau rank correlation metric. MERRA-2 Kendall Tau rank correlations are > 0.6 for dust above 2 km and for BC/combustion aerosol layers between 1.5-7 km. FLEXPART rank correlations are > 0.6 above ~3 km for dust, and up to around 5 km for BC (and sometimes OC). These findings indicate that at these altitudes and scales MERRA-2 and FLEXPART dust and BC have similar large-scale patterns as CALIPSO dust and combustion aerosols, respectively, although a sizeable amount of variability is still unaccounted for.

Figure 3 also indicates that correlations between the MERRA-2 and CALIPSO dust are increasingly poor near the surface. This observation could be due either to the increasing influence of factors affecting aerosol optical properties not represented by dust concentrations alone (e.g., pollution or sea salt mixed with dust), or errors in CALIPSO or MERRA-2/FLEXPART at these altitudes. Figure 3 also shows how the correlations change if "polluted dust" is excluded from the CALIPSO dust and combustion aerosol classification. Removing polluted dust did not have a large effect on relationships with

combustion aerosols, but it reduced nighttime dust correlations for both MERRA-2 and FLEXPART at most altitudes, and substantially so for MERRA-2 near the surface. We conclude that polluted dust is an important contributor to total dust loads during polar night.

Daytime MERRA-2 and FLEXPART dust (shown for comparison in Figure 3, white backgrounds) appear to be slightly less correlated with CALIPSO at some altitudes but similarly or maybe even slightly more correlated for combustion aerosols. However, because CALIPSO detects fewer aerosol layers during the daytime, the remainder of our discussion is focused on nighttime data when the CALIPSO comparison data are of higher quality.

Figure 4 shows where MERRA-2/FLEXPART and CALIPSO agree most on aerosol distributions based on differences in Z-scores between the products. For combustion aerosols, 87% and 89% of the respective MERRA-2 and FLEXPART output have Z-score differences of < 1.0 based on BC levels (Fig. 4). However, relative to CALIPSO, MERRA-2 BC and OC concentrations have larger discrepancies than FLEXPART over the North Atlantic and Eurasia in the lower troposphere (0.2 to 2 km).

There is a notable discrepancy between MERRA-2 dust and CALIPSO dust layer presence over Europe that is not seen for FLEXPART (Fig. 4). Therefore, it is possible that MERRA-2 is overestimating long-range dust transport over Europe, which is contributing to the lower MERRA-2 correlation with CALIPSO dust at these altitudes (Fig. 3). In contrast, some locations with high local dust sources (such as Iceland and the Canadian Archipelago) seem to have high FLEXPART to CALIPSO dust discrepancies that are not observed for MERRA-2 (Fig. 4). These locations match tightly with local emissions in the FLEXPART model (see Figure 2 in Groot Zwaaftink et al., 2016), suggesting that local dust contributions to the Arctic atmosphere may be too high, which might contribute to the similarly lower correlations for FLEXPART with CALIPSO at these lower altitudes (Fig. 3). However, it should be noted that if CALIPSO uniformly underestimates near-surface dust for some reason, that might produce similar patterns. Over Siberia between 0.2-2 km, where RHi values tended to be high (Fig. 2), there was agreement between CALIPSO, MERRA-2 and FLEXPART that dust levels were elevated (Fig. 1). The Z-score differences in this area were generally < 1.5 (Fig. 4).

## 3.4 The importance of local dust

There are only a few locations where Arctic dust aerosol concentrations have been directly measured over long time periods, but we know that local dust emissions can be important aerosol sources at specific sites, such as near receding glaciers (Bullard and Mockford, 2018; Gassó et al., 2018). In Figure 5, we compare long-term ground-based dust observations at Stórhöfði, Iceland (a site of large local dust emissions) from Prospero et al. (2012) to dust concentrations during the study period from MERRA-2 and FLEXPART.

Observations were set up to study dust from remote sources and therefore samples were only taken during wind directions south to west, while this selection is not done in modelled data. One could therefore expect a better match with

MERRA-2, which only includes remote sources, and overestimations by FLEXPART, which also includes local sources. Clearly both models leave something to be desired when it comes to matching the observations at this site. FLEXPART most of the time overestimates dust concentrations, in line with the CALIPSO observations from Figure 4. In contrast, MERRA-2 underestimates dust throughout the year. Generally, the mean MERRA-2 dust concentration is closer than FLEXPART to the observed mean from July through March. However, FLEXPART dust variability is high enough to include the occasional extreme dust event observed episodically at this site, i.e., when dust concentrations exceed 100 μg dust m$^{-3}$ (Prospero et al., 2012). These events are most common in the spring when local emissions are highest. In contrast, a dust concentration of 100 μg m$^{-3}$ is many standard deviations outside the mean of the MERRA-2 values at this site. As such, it appears that MERRA-2 is less able to capture dust extremes than FLEXPART at this site, likely because it currently does not account for local dust sources. Wu et al. (2020) showed similar data at Stórhöfði (Heimaey) for MERRA-2, and showed that ground dust observations at Alert, Greenland were also underpredicted. However, they found that MERRA-2 tended to overpredict dust aerosol optical depth in many remote regions of the Arctic compared to Arctic satellite dust optical depths from CALIPSO. The underprediction of dust at the Icelandic and Greenland sites despite a potential overprediction bias at most other sites further underscores the need to improve the relative spatial distributions of dust in the reanalysis product by adding local dust emission sources.

## 4 Conclusions

Aerosol models are often used in aerosol-cloud interaction studies over the climatically sensitive Arctic region. Our goals in this study were to: 1) to better understand the strengths and limitations of the Arctic MERRA-2 and FLEXPART dust and combustion aerosol products and to suggest where they might be improved, and 2) to combine satellite, reanalysis, and model products (each with their own limitations and biases) to learn more about where there is high confidence in the concentrations of aerosols, and how that relates to meteorological conditions conducive to the formation of diamond dust.

To summarize the model validation portion of the study, we found that both MERRA-2 and FLEXPART can provide useful information for aerosol studies in the Arctic region at high altitudes, where clouds and other factors often make it difficult to observe aerosols from space and where *in situ* observations are scarce. They each predict the vertical distribution of combustion aerosols with relatively high confidence, despite some likely CALIPSO combustion aerosol subtyping artifacts over the Arctic Ocean. They also provide useful information on large-scale patterns of dust above ~2 km, although a large amount of variability is still unaccounted for, especially near the surface.

When using the MERRA-2 product to estimate dust aerosols, it would be useful to keep in mind that CALIPSO overestimates dust concentrations substantially in conditions where diamond dust is present. We estimate that more than a third of lower troposphere wintertime Arctic CALIPSO mineral dust observations could actually be diamond dust instead of mineral dust. This observation is supported by other studies (e.g., Di Biagio et al., 2018) and is evidenced by an ~60% increase in CALIPSO dust classifications in conditions conducive to diamond dust formation (i.e., where $RH_i > 100\%$) and a substantial

drop in the correlation between CALIPSO dust layer presence and MERRA-2 and FLEXPART dust concentrations from ~0.6 at certain altitudes to near zero. Moreover, local dust sources in MERRA-2 are lacking, leading to underestimates in dust presence based on comparisons to CALIPSO and ground observations, especially in boreal spring. Adding local dust emission sources into MERRA-2 should improve the distribution of dust in the Arctic for this product. In contrast, FLEXPART includes

local dust sources and may capture the high dust variability from local sources, but it appears to currently overestimate these contributions during winter.

To summarize the second part of the study, we provide evidence that the impacts of mineral dust and combustion aerosols on Arctic ice particle formation may be higher over Siberia than over other Arctic regions. CALIPSO, MERRA-2, and FLEXPART show agreement that Siberia has high wintertime mineral dust and combustion aerosol levels, likely from long-

range transport. Moreover, this region is very likely to experience conditions conducive to diamond dust formation frequently; during winter, conditions favourable to diamond dust formation occurred up to 60% of the time in the altitude ranges we assessed. Such conditions hardly ever occur during summer and are not expected to contribute to CALIPSO dust aerosol uncertainties much during this season. This finding suggests that aerosol effects on clouds in a warming Arctic are changing, as the temporal window and locations where diamond dust and heterogenous freezing can occur is shrinking, even as local

mineral dust sources may be increasing due to retreating glaciers (Prospero et al., 2012).

This study focused on dust and combustion aerosols. However, other recent studies indicate that marine aerosols may be particularly important for the Arctic ice nucleating particle budget (e.g., Carlsen and David (2022)). For example, Porter et al. (2022) found that biogenic particles may dominate the ice nucleating particle budget even over the Russian coast where we found that the dust aerosol impacts are likely to be highest. Thus, aerosol sources besides dust may enhance aerosol impacts

on clouds over this region even further and could have a larger impact on clouds across the Arctic region than the dust and combustion aerosols of focus in this study, particularly at times when the surface is not covered with snow or ice.

Lastly, the study helps us identify several areas where future research is likely to be particularly fruitful or helpful.

1. Models can help identify where changing temperatures, moisture fluxes and aerosol types and concentrations are most likely to impact future aerosol homogeneous and deposition ice nucleation pathways.

2. Non-CALIPSO platforms (such as Raman lidar) can help better distinguish diamond dust from mineral dust aerosol over Siberia during winter, which will enable better validation of modelled wintertime aerosols, and better overall assessment of mineral dust and combustion aerosol impacts on freezing processes over this region.

3. Aircraft, ground, and ship data can help validate models of aerosols from different sources over remote Arctic

locations, and can better assess their contributions to INP budgets. Based on our analysis, some regions of particular interest would be places where marine and local dust emissions are thought to be strongest.

4.  Laboratory, satellite, and aircraft data can help better assess the extent to which diamond dust forms through pre-activation processes, which will help elucidate how often diamond dust and ice nucleation actually occur at locations with $RH_i$ values < 100%.


**Code and data availability**

See AIRS project (2019) for the AIRS data, Winker (2018) for the CALIPSO data, Amante and Eakins (2009) for the ETOPO
data, Zamora et al. (2022a, b, c) for the FLEXPART output, and GMAO (Global Modeling and Assimilation Office (GMAO),
2015a, b) for the MERRA-2 output.

**Author contributions**

LZ, RK, and KH designed the experiments and LZ carried them out. NE and CGZ ran FLEXPART and provided simulation
output. LZ prepared the manuscript with contributions from all co-authors.

**Competing interests**

The authors declare that they have no conflict of interest.

**Acknowledgements**

Resources supporting this work were provided by the NASA High-End Computing (HEC) Program through the NASA Center
for Climate Simulation (NCCS) at Goddard Space Flight Center. Data from Figure 1 were produced with the Giovanni online
data system, developed and maintained by the NASA GES DISC. LZ and RK would like to thank H. Bian, J. Creamean, A.
Darmenov, J. Lee and the ARCSIX science writing team for helpful discussions. LZ and the contributions of RK were
supported by the NASA Aerosol-Cloud Modeling and Analysis Program (Grant 80NSSC19K0978) under Richard Eckman.
NE was supported by the COMBAT (Quantification of Global Ammonia Sources constrained by a Bayesian Inversion
Technique) project funded by NFR's ROMFORSK – Program for romforskning of the Research Council of Norway (Project
ID: 275407).

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

**Table 1: Conditions under which diamond dust can form over the Arctic. Heterogeneous ice formation also requires the presence of ice nuclei.**

| Formation mechanism | T ($^{o}$C) | RHi (%) |
|---|---|---|
| Homogeneous formation[a] | < -35 $^{o}$C | > 140% |
| Heterogeneous formation: Deposition ice nucleation[b] | < 0 $^{o}$C | > 100% |
| Heterogeneous formation: homogeneous freezing of preactivated water-containing aerosol pores[c,d] | < -38 $^{o}$C | < 100% |

a) Koop et al. (2000); b) Kanji et al. (2017); c, d) (Marcolli, 2014, 2017)

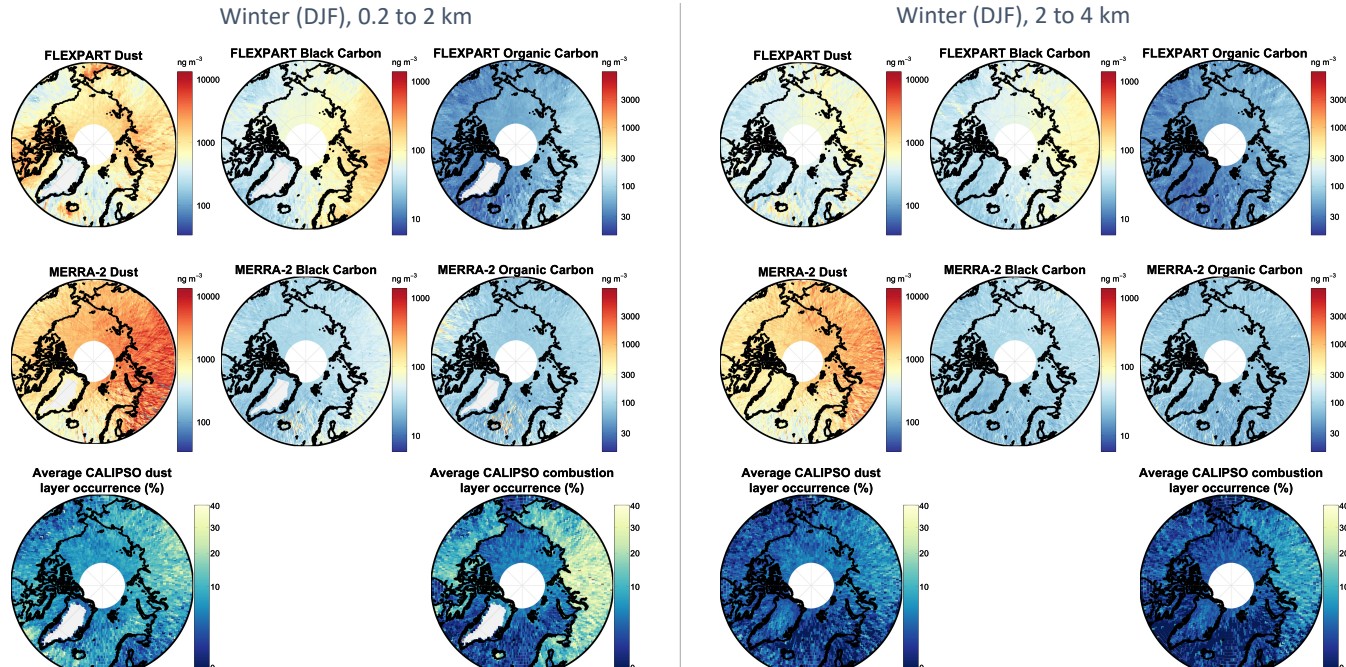


**Figure 1: The percent of the time that CALIPSO dust and combustion aerosols occur in wintertime cloud-free profiles below 4 km at two different altitude levels, and the corresponding distributions of MERRA-2 and FLEXPART submicron mineral dust, black carbon, and organic carbon. There is agreement that dust (an efficient source of ice nucleating particles) is high over Siberia. Data at higher altitudes are shown in Fig. S1.**


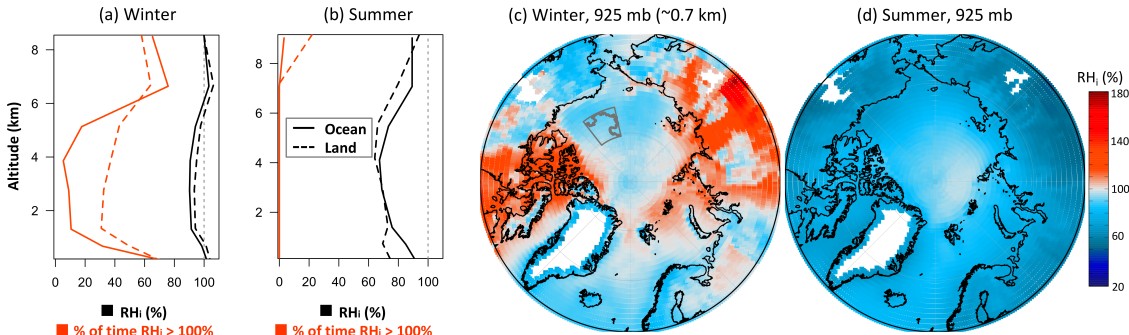

**Figure 2:** Diamond dust is unlikely to occur during the summer and is more likely to occur during winter, especially over the Canadian archipelago and over Siberia, based on seasonally averaged RH$_i$ distributions for the winter (DJF) and summer (JJA) between 2008-2016. On the left are the vertical average RH$_i$ profiles over the Arctic Ocean (solid lines, including areas covered by sea ice) and land regions (dashed lines) for a) winter, and b) summer. For reference, also shown are the percentages of these regions in which RH$_i$ is > 100% (red lines). On the right are RH$_i$ distributions at the 925 mb isobar for c) winter, and d) summer. Also shown in dark grey in c) is the SHEBA cruise track and the region surrounding it. Diamond dust is most likely to occur where RH$_i$ values are > 100% (red regions).

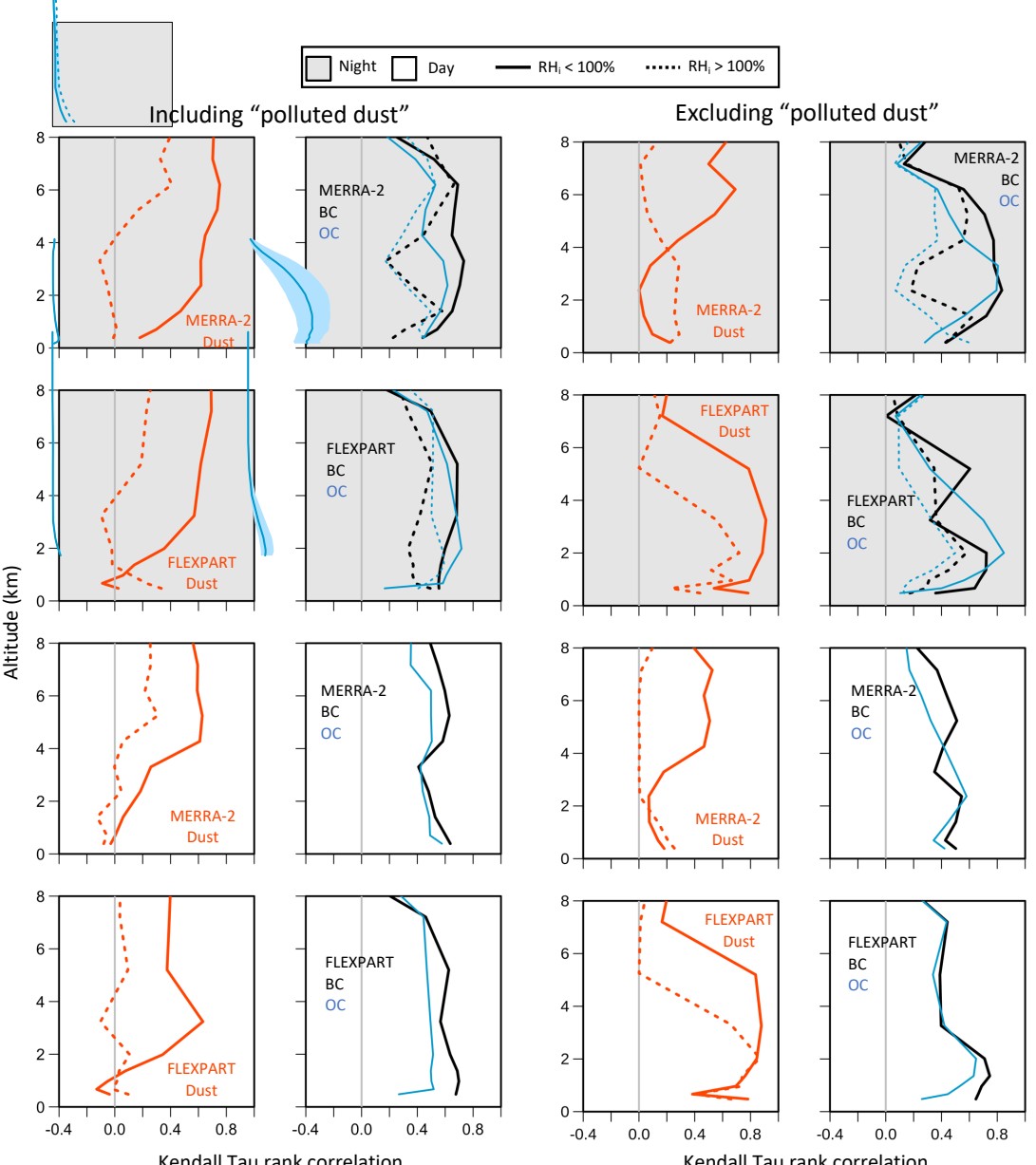


**Figure 3: The Arctic area-averaged Kendall Tau rank correlation between mean CALIPSO aerosol layer fraction for dust (far left) and combustion aerosols (second column) with MERRA-2 and FLEXPART dust and black carbon, (BC, black) and organic carbon (OC, blue) concentrations. The right two columns show the same information but with "polluted dust" excluded from both CALIPSO dust and combustion aerosol groups. Data are shown for nighttime (SZA>90°) and daytime (SZA<90°) conditions (grey and white backgrounds, respectively), and for conditions conducive to diamond dust formation (dashed lines). Model bins with < 30 observations were excluded in the averaging, which excludes < 1% of total model bins. Comparisons were made from 20 x 6 degree bin averages on cloud-free days with the goal of capturing large-scale features.**


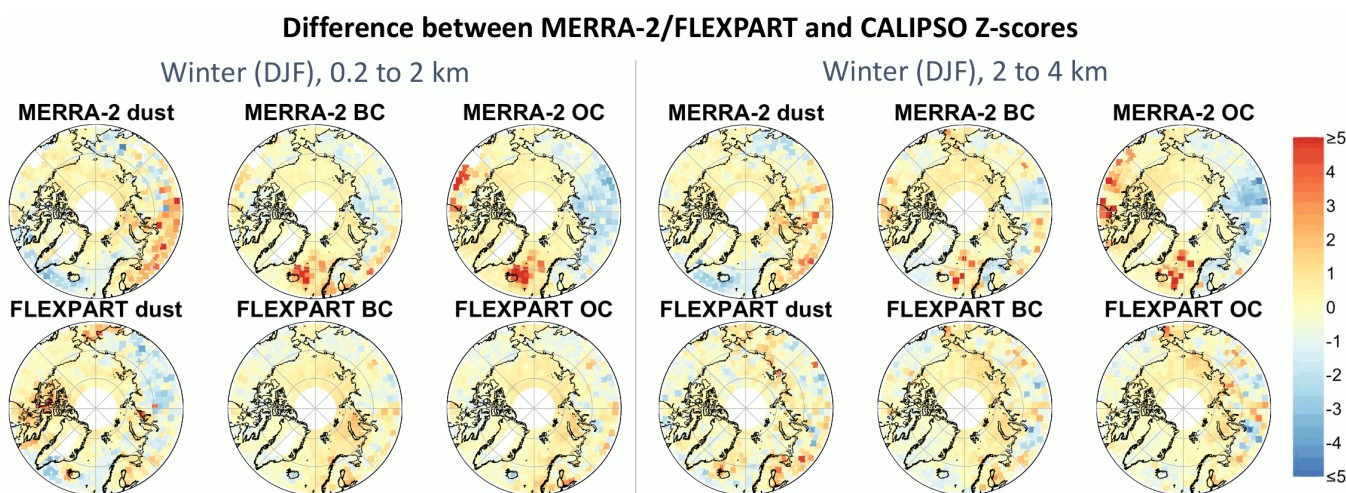

**Figure 4: Left column: The difference in Z-scores (i.e., standard deviations from their respective means) between average MERRA-2/FLEXPART dust concentrations and CALIPSO dust aerosol layer presence below 4 km. The Z-score differences are shown for nighttime winter (DJF) periods in non-diamond-dust conditions. Middle and right columns: as in the left column, except CALIPSO combustion aerosols are compared to MERRA-2/FLEXPART black carbon (BC) and organic carbon (OC) concentrations. Locations with < 30 observations are excluded in the above plots. Data from higher altitudes are shown in Fig. S4.**


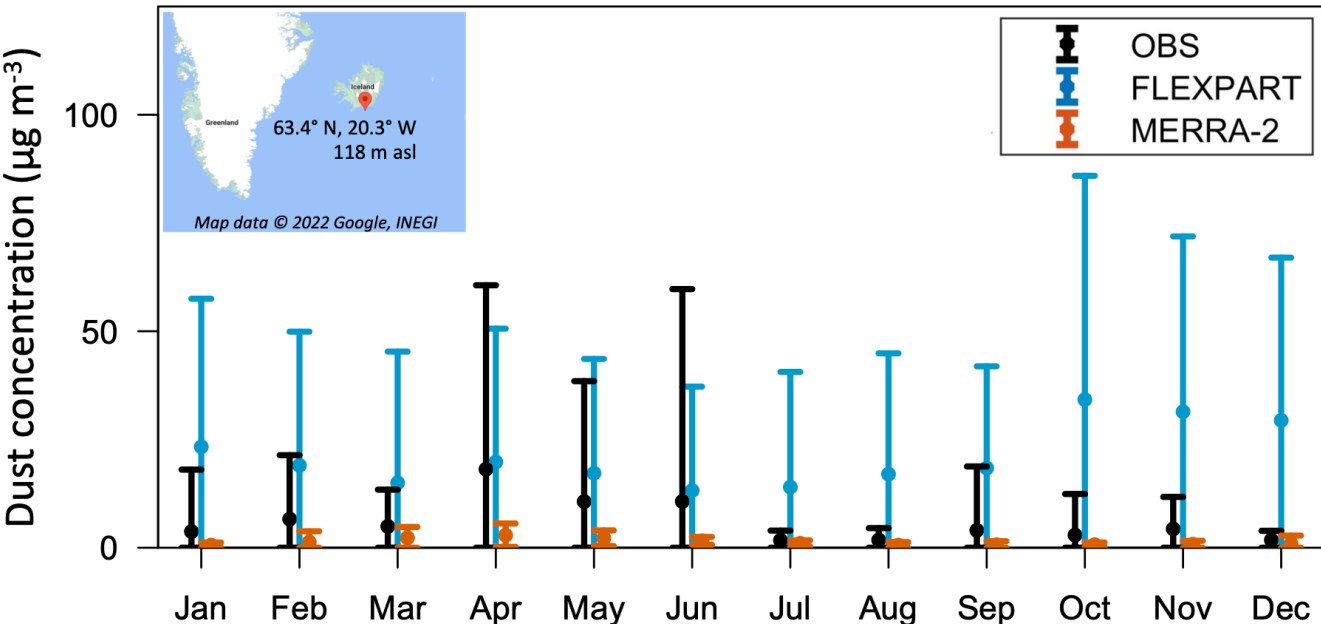

**Figure 5: We compared long-term ground-based mineral dust observations at an Icelandic site with large local dust emissions (Stórhöfði, Heimaey) from Prospero et al. (2012) to dust concentrations during the 2008-2015 study period from MERRA-2 and FLEXPART. The whiskers show mean and standard deviation. MERRA-2 (brown) substantially underestimated observed mineral dust concentrations compared to observations (black), particularly in the spring when local emissions are highest. Adding local dust emission sources into MERRA-2 should improve the product.**