# Peer review of "Comparisons between the distributions of dust and combustion aerosols in MERRA-2, FLEXPART, and CALIPSO and implications for deposition freezing over wintertime Siberia"

_Atmospheric Chemistry and Physics, 2022_

## Referee Comment (RC1)

In this study, the authors compared mineral dust and combustion aerosol distributions from CALIPSO, MERRA-2, and FLEXPART model in the Arctic during 2008-2015. They found that both MERRA-2 and FLEXPART predict the vertical distribution of combustion aerosols with relatively high confidence, as does FLEXPART for mineral dust. MERRA-2 and FLEXPART have substantially higher false negative rates for mineral dust in conditions favouring diamond dust formation. All three products predicted that wintertime dust and combustion aerosols occur most frequently over the same Siberian regions where diamond dust is most common in the winter. The manuscript is well written, and results are clearly presented. I have a few comments for the authors to consider.

General Comments:

The dust size in diameter in MERRA-2 (GOCART aerosol model) should be 0.2-2, 2-3.6, 3.6-6, 6-12, and 12-20 μm. Many papers have cited incorrectly. Please make it clear whether the size ranges are for radius or diameter (both MERRA-2 and FLEXPART), and it is better to be consistent to use diameter as FLEXPART (the numbers appear to be for diameter). For the grouping of the bins, bin 1 contains not only submicron but also supermicron dust. If you want to exclude dust with diameter larger than 10 μm, you may need to use bins 2-4. Or you can use all bins. MERRA-2 has the size up to 20 μm as FLEXPART does. Please make sure that the size bins for MERRA-2 are correctly used and referred throughout the paper.

The authors focused on qualitative comparisons of aerosol presence information between CALIPSO and MERRA-2/FLEXPART. I'm a little confused about the methodology, especially for the introduction/explanation of Table 2. It would be nice if the authors could explain more in details. What does the tested range of minimum concentrations in Table 2 mean? Ranges like "BC: >41 to >100 ng m$^{-3}$" are confusing. Different vertical levels have different threshold values? Does that minimum range correspond to 67-92.5% quantile? What does 67-92.5% quantile mean? Larger than 67% of the values to 92.5% of the values? Why MERRA-2 and FLEXPART use the same quantile range? How are the range of minimum concentrations used for the calculations of FN and FP rates? How would you determine the amount of dust, BC, and OC to be detectable with CALIPSO? Is it ideal to use a simulator? "Polluted dust" was included in both "dust" and "pollution" group. How much could this affect the FP and FN rates?

Specific comments:

Line 128-129, the assimilation of aerosol data in MERRA-2 outside the Arctic may greatly increase the transport. For mineral dust, the lower to middle troposphere may be affected more by local sources. The middle to upper troposphere may be affected more by the transport.

Line 204-205, it is a little confusing. Is it CALIPSO vs MERRA-2/FLEXPART or dust vs combustion aerosol? Consider rephrasing it?

Figure 1, the panels seem to be small comparing to the large white space between them. Please consider adjusting them. Try to make the panels larger and compact.

Figure 3, "Both models can predict the … that near-surface dust sources may be underestimated" These should better be discussed in the main text instead of in the figure caption.

Figure 4, there seems to be too much white space, like Figure 1, try to make the panels larger and compact. For the colorbar, usually one color indicates a value range. The red and blue color in the colorbar indicate >100% and <0% values? Please change it if necessary.

---

## Referee Comment (RC2)

Review of "Comparisons between the distributions of dust and combustion aerosols in MERRA-2, FLEXPART and CALIPSO and implications for deposition freezing over wintertime Siberia" by L. M. Zamora, R. A. Kahn, N. Evangeliou and C. D. Groot Zwaaftink

In this well-written manuscript, the authors offer several interesting comparisons between models (MERRA2 and FLEXPART) and observations (CALIPSO). But because they confine themselves to "qualitative comparisons", these comparisons are somewhat less informative than I had hoped when I first agreed to review the paper. I was particularly disappointed by the lack of a quantitative assessment of CALIPSO's propensity to misclassify diamond dust as mineral dust. Nevertheless, I believe the authors' results will be helpful in understanding the limits of both measurements and models in the Arctic environment. That being the case, I'm happy to recommend publication once the authors address the comments embedded in the annotated version of their manuscript appended below.

[revised manuscript text omitted]

---

## Author Comment (AC1)

**General response:**

We would like to thank the Reviewers for their thoughtful comments and efforts towards improving the manuscript. We particularly appreciate the detailed review of the MERRA-2 and CALIPSO methodology, which allowed us to revise and clarify several aspects of the methods. We believe that these changes strengthen the robustness of our results, without altering the main message of the paper.

We start by listing significant changes and follow with a point-by-point response to each Reviewer.

An important update is that we have improved our approach to assessing model performance, focusing on correlations and differences in aerosol distributions (Z-scores), and emphasizing larger scale patterns as opposed to case-by-case (small-scale) comparisons. This method is more intuitive and more quantitative, as suggested by the Referees. One problem with the old approach is that, because CALIPSO aerosol cases in the Arctic are so comparatively rare (e.g., dust layers only occur <5% of the time), false positive rates are always approximately equal to the pre-defined aerosol quantile cutoff values for what might be considered an aerosol layer, and are thus not a very meaningful scientific metric. Moreover, we found a bug in the code that when fixed, resulted in substantially higher false negative and positive rates than were previously reported, making it apparent that an analysis of large-scale aerosol trends would be more appropriate. We updated Section 2.2 "Comparisons between MERRA-2, FLEXPART, and CALIPSO" to describe the new method used to assess the model performance, as well as Section 3.3 "Model aerosol evaluation." Model performance by the new metrics supports the original major findings of the paper. On a related note, Dr. Klaus Huebert (a statistician who helped us develop the new method for assessing model performance) has been added to the co-author list.

Other important changes are listed below:

- Fig. 3 now includes a panel showing how the results would change if "polluted dust" were excluded from both the CALIPSO dust and combustion aerosol groups, to address a question both referees asked.
- The new Fig. S2 shows how much more frequently CALIPSO observes mineral dust in conditions that favor diamond dust formation, as one of the referees was interested in how diamond dust could affect errors in CALIPSO aerosol subtyping.
- Fig. S3 is the same as the new Figure 3, except that it shows correlations with Pearson correlation metric, which may be more familiar to some readers than the Kendall Tau metric used in Figure 3.
- Table 2 has been removed, as it related to the method for deriving false negative and positive rates that is no longer used.

Additional changes to the manuscript and the Supplement are detailed in the following point-by-point response.

**Reviewer #1:**

Comments:

In this study, the authors compared mineral dust and combustion aerosol distributions from CALIPSO, MERRA-2, and FLEXPART model in the Arctic during 2008-2015. They found that both MERRA-2 and

FLEXPART predict the vertical distribution of combustion aerosols with relatively high confidence, as does FLEXPART for mineral dust. MERRA-2 and FLEXPART have substantially higher false negative rates for mineral dust in conditions favouring diamond dust formation. All three products predicted that wintertime dust and combustion aerosols occur most frequently over the same Siberian regions where diamond dust is most common in the winter. The manuscript is well written, and results are clearly presented. I have a few comments for the authors to consider.

We thank Reviewer #1 for the very helpful review. Below we provide a point-by-point response to each of the comments.

General Comments:
The dust size in diameter in MERRA-2 (GOCART aerosol model) should be 0.2-2, 2-3.6, 3.6-6, 6-12, and 12-20 µm. Many papers have cited incorrectly. Please make it clear whether the size ranges are for radius or diameter (both MERRA-2 and FLEXPART), and it is better to be consistent to use diameter as FLEXPART (the numbers appear to be for diameter).

As suggested, MERRA-2 dust sizes (which had previously been listed in radius) are now listed in diameter and are consistent with the values the reviewer listed above. We also specify that the FLEXPART dust particle sizes are listed in diameter in section 2.1.4.

For the grouping of the bins, bin 1 contains not only submicron but also supermicron dust. If you want to exclude dust with diameter larger than 10 µm, you may need to use bins 2-4. Or you can use all bins. MERRA-2 has the size up to 20 µm as FLEXPART does. Please make sure that the size bins for MERRA-2 are correctly used and referred throughout the paper.

Thank you for pointing that out that error. We re-ran the Figures using all bins up to the size of 20 µm so that the dust comparison is as similar as possible between MERRA-2 and FLEXPART, but this does not change the main findings of the study.

The authors focused on qualitative comparisons of aerosol presence information between CALIPSO and MERRA-2/FLEXPART. I'm a little confused about the methodology, especially for the introduction/ explanation of Table 2. It would be nice if the authors could explain more in details. What does the tested range of minimum concentrations in Table 2 mean? Ranges like "BC: > 41 to > 100 ng m$^{-3}$" are confusing. Different vertical levels have different threshold values? Does that minimum range correspond to 67-92.5% quantile? What does 67-92.5% quantile mean? Larger than 67% of the values to 92.5% of the values? Why MERRA-2 and FLEXPART use the same quantile range? How are the range of minimum concentrations used for the calculations of FN and FP rates? How would you determine the amount of dust, BC, and OC to be detectable with CALIPSO? Is it ideal to use a simulator?

Both referees found this method confusing, and due to the issues described in the General Response above, we are no longer using the false positive/negative rate method. The original Table 2 and associated text have been removed.

"Polluted dust" was included in both "dust" and "pollution" group. How much could this affect the FP and FN rates?

Both referees were interested in this question. To address it, we added a panel to Fig. 3 that shows how the results change if "polluted dust" is excluded from both the CALIPSO dust and combustion aerosol groups.

In addition, we added the following text to the methods in Section 2.2:

"Any CALIPSO aerosol layer in cloud-free conditions that was classified as either "dust," "polluted dust," or "dusty marine" was included in a larger dust group for comparison to other datasets. Any CALIPSO aerosol layer classified as either "polluted continental/smoke", "polluted dust," or "elevated smoke" was included in a larger combustion aerosol group. "Polluted dust" was included in both the larger dust and pollution groupings, reflecting the fact that aerosols of both types can be mixed in the atmosphere. Polluted dust made up 45% of the dust cases, and 39% of the combustion cases, and so there is significant overlap between the two groups. We discuss later how removing the "polluted dust" component of the combustion aerosol group effects the results."

And we added the following text to the results in Section 3.3:

"Figure 3 also shows how the correlations change if "polluted dust" is excluded from the CALIPSO dust and combustion aerosol classification. Removing polluted dust did not have a large effect on relationships with combustion aerosols, but it reduced nighttime dust correlations for both MERRA-2 and FLEXPART at most altitudes, and substantially so for MERRA-2 near the surface. We conclude that polluted dust is an important contributor to total dust loads during this time of year."

Specific comments:
Line 128-129, the assimilation of aerosol data in MERRA-2 outside the Arctic may greatly increase the transport. For mineral dust, the lower to middle troposphere may be affected more by local sources. The middle to upper troposphere may be affected more by the transport.

We updated the text as follows:

"During the time period of this study, MERRA-2 assimilated aerosol information from the Aerosol Robotic Network (AERONET), Multi-angle Imaging SpectroRadiometer (MISR), and Moderate Resolution Imaging Spectroradiometer (MODIS) instruments (Randles et al., 2017). Data from these instruments were assimilated throughout the year in the subarctic, and a fraction of these aerosols were then transported into the Arctic. During the daytime, MERRA-2 also assimilates some limited Arctic aerosol data in non-cloudy conditions. However, those Arctic data sources are unavailable during polar night. As a result, MERRA-2 aerosol output is more model-driven during polar night. Uncertainties may be largest near the surface, which is more impacted by local sources, than in the middle and upper troposphere, which are more influenced by transported aerosols."

Line 204-205, it is a little confusing. Is it CALIPSO vs MERRA-2/FLEXPART or dust vs combustion aerosol? Consider rephrasing it?

As suggested, this has been rephrased for clarity.

Figure 1, the panels seem to be small comparing to the large white space between them. Please consider adjusting them. Try to make the panels larger and compact.

Done.

Figure 3, "Both models can predict the ... that near-surface dust sources may be underestimated" These should better be discussed in the main text instead of in the figure caption.

As suggested, we have removed this information from the figure caption. Interpretation of the figure is now primarily discussed in section 3.3.

Figure 4, there seems to be too much white space, like Figure 1, try to make the panels larger and compact. For the colorbar, usually one color indicates a value range. The red and blue color in the colorbar indicate > 100% and < 0% values? Please change it if necessary.

Done.

**Reviewer #2:**

Comments:

In this well-written manuscript, the authors offer several interesting comparisons between models (MERRA2 and FLEXPART) and observations (CALIPSO). But because they confine themselves to "qualitative comparisons", these comparisons are somewhat less informative than I had hoped when I first agreed to review the paper. I was particularly disappointed by the lack of a quantitative assessment of CALIPSO's propensity to misclassify diamond dust as mineral dust. Nevertheless, I believe the authors' results will be helpful in understanding the limits of both measurements and models in the Arctic environment. That being the case, I'm happy to recommend publication once the authors address the comments embedded in the annotated version of their manuscript appended below.

We thank the reviewer for their thoughtful and thorough comments, which helped to clarify and improve this paper. At the reviewer's suggestion, we made the overall analysis more quantitative and less qualitative in two ways:

1) We changed the method for assessing model performance, from a qualitative analysis of false positive and negative rates to a quantitative analysis of correlations and differences in Z-scores (see the new section 2.2).

2) At the referee's suggestion, we also added in a more quantitative assessment of how frequently CALIPSO can misclassify diamond dust as mineral dust in the supplement (see Figure S2).

Specific Comments:

Abstract: "We compared Arctic 2008-2015 mineral dust and combustion aerosol distributions from the Cloud-Aerosol Lidar and Infrared Pathfinder Satellite Observation (CALIPSO) satellite, the Modern- Era Retrospective analysis for Research and Applications, Version 2 (MERRA-2) reanalysis products, and the FLEX-ible PARTicle (FLEXPART) model. Based on Atmospheric Infrared Sounder (AIRS) satellite meteorological data, diamond dust may occur up to 60% of the time in winter, but it hardly ever occurs in summer."

question 1: what are the spatial and temporal dimensions of the AIRS data used in this scheme?

We re-worded the abstract to say, "Based on **coincident, seasonal** Atmospheric Infrared Sounder (AIRS) **Arctic** satellite meteorological data,…"

The 60% statement refers to Figure 2a, which is based on seasonally averaged AIRS data during the study period from the ascending and descending orbits. However, as is later described in section 2.1.2, AIRS data were also taken concurrently with the CALIPSO data to determine whether each individual profile was present in conditions that could support diamond dust formation (defined as $RH_i > 100\%$). AIRS data are taken within minutes of the CALIPSO data on the A-train.

The spatial resolutions of the AIRS datasets are discussed in further detail in section 2.1.2.

"However, CALIPSO can miss aerosols that are … in the 200 m immediately above the surface where local marine and terrestrial emission concentrations are highest, …." please provide a literature reference for this assertion.

We now cite Winker, D.M., J.L. Tackett, B.J, Getzewich, Z. Liu, M.A. Vaughan, and R.R. Rogers, 2013. The global 3-D distribution of tropospheric aerosols as characterized by CALIOP. Atmos. Chem. Phys., 13, 3345–3361, doi:10.5194/acp-13-3345-2013.

asl and agl: are these acronyms so widely understood and accepted that they can go without being formally defined in an EGU publication? and, since they're acronyms, should they be capitalized?

We now define and capitalize ASL and AGL in the text.

"For this analysis, CALIPSO aerosol data were used only from cloud-free profiles where CAD scores ranged between -100 and -30, to exclude clouds and very low-confidence aerosol layers. CALIPSO aerosol layer properties have higher uncertainties during the daytime, especially over bright sea ice. As mentioned previously, CALIPSO may also miss dilute Arctic aerosols, even at night, when lidar sensitivity is higher (Zamora et al., 2017). Moreover, aerosol subtype designation (Omar et al., 2009; Kim et al., 2018) can be subject to other errors as well. In general, aerosol type is more difficult to discern in aerosol mixtures (Zeng et al., 2021). Marine aerosols are not allowed in over-land retrievals, but may still be present, at least near coastal locations (Kanitz et al., 2014). Also, desiccated marine aerosols might be misclassified as dust or polluted aerosol at relative humidity below 60-70% (Ferrare et al., 2020), and pollution aerosols can be misclassified as marine aerosols (Di Biagio et al., 2018)." this is a nice assessment of the potential pitfalls in the CALIOP aerosol typing scheme.

Thank you.

"Details of the MERRA-2 Arctic aerosol distributions have been discussed previously (Wu et al., 2020; Lee et al., 2020; Sitnov et al., 2020)." what distinction are the authors drawing between "discussed" and "evaluated"?

The sentence has been clarified as follows:
"Only a few studies have evaluated MERRA-2 Arctic aerosol distributions, mainly using MODIS aerosol optical depth data and ground-based observations (e.g., Wu et al. (2020), Lee et al. (2020) and Sitnov et al. (2020))."

"We did not analyse dust with sizes > 10 μm for easier comparison with MERRA-2, which only assessed particles up to this size." I wonder what's missed by taking this approach? See Drakkili et al., ACPD 2022, https://doi.org/10.5194/acp-2022-94: "In this study we extend the particle size range acknowledged in WRF-GOCART-AFWA transport code, to include particles with diameters up to 100 μm. The evaluation against airborne in situ observations of the size distribution shows that larger particles, are underestimated, both above their sources and far from them. This suggests that there are atmospheric processes that are not taken into account in the model simulations."

Reviewer #1 pointed out that we actually made an error in the dust size classes from MERRA-2, which go up to 20 μm instead of 10 μm. We fixed this error and redid the figures and tables so that now these larger size classes are included.

We now also reference the above paper as follows:
"Note that as with many other dust models, particles with diameters >20 μm are not assessed. Such larger dust particles are comparatively rare but are observed in the field (Weinzierl et al., 2017; Drakaki et al., 2022) and so their exclusion could lead to an underestimate of *in situ* dust concentrations, especially near local sources."

"...various observations suggest that FLEXPART BC can proxy strong, CALIPSO-detectable aerosol layers…" I've always understood the word "proxy" as either a noun or an adjective.  This is my first-ever encounter with it being used as a verb.
https://www.merriam-webster.com/dictionary/proxy
https://dictionary.cambridge.org/us/dictionary/english/proxy
https://www.collinsdictionary.com/us/dictionary/english/proxy
https://www.britannica.com/dictionary/proxy

We changed the sentence to read, "various observations suggest that FLEXPART BC can **be a** proxy **for** strong, CALIPSO-detectable aerosol layers"

"In order to meet our goal of better understanding the strengths and limitations of MERRA-2 and FLEXPART dust and combustion aerosol products, we focused primarily on environmental conditions where the CALIPSO satellite product does best. These conditions include cloud-free, nighttime cases when diamond dust does not occur, at altitudes > 200 m over the surface." do the authors mean "include" or "are limited to"; if the former, please enumerate the other conditions that are also included.

The paragraph has been changed as follows:

"We also focused primarily on environmental conditions where the CALIPSO satellite product does best. These conditions include cloud-free, nighttime cases when diamond dust does not occur, at altitudes > 200 m over the surface. **However, for comparison, correlations were also analysed separately during daytime (defined as when the solar zenith angle (SZA) is < 90º) when the lidar signal-to-noise ratio is smaller, and when RHi is > 100%, when potential aerosol type errors from diamond dust might be highest**."

Table 2 caption: "The MERRA-2 and FLEXPART aerosol layer definitions are based on a range of potential minimum model concentrations, above which CALIPSO aerosols are assumed to be detectable,…" what's the basis for this assumption?  are these detection assumptions consistent with the minimum detectable backscatter predictions described in the CALIPSO layer detection ATBD?

Table 2 related to the previous method for assessing model performance and is no longer included in the paper. The new methods no longer require assumptions about the range of aerosol concentrations assumed to be detectable.

on lines 212-213 the authors make this statement:
'Because this analysis focuses on aerosol concentration quantile values rather than on absolute concentrations, the aerosol concentration thresholds for determining a "concentrated" aerosol layer are different between MERRA-2 and FLEXPART (see Table 2).' Having this statement appear before table 2 would have minimized a whole bunch of head-scratching on my part.  I suspect this would also be true for other readers of this paper.

This text is related to the previous method for assessing model performance and is no longer included in the paper.

"[b]Based on the 2008 67-92.5% quantile of aerosol concentrations from 0.25-6 km." what is the source for the number? what is the rationale for choosing this range? please explain further.

This text is related to the previous method for assessing model performance and is no longer included in the paper.

"To derive a parameter such as aerosol concentration from CALIPSO, one would need to make speculative assumptions about the lidar ratio and the extinction cross section of particles, which is beyond the scope of this paper." determining whether CALIOP will detect a vertically distributed target requires more than just an estimate of the aerosol concentration. estimates of the aerosol backscattering and extinction cross-sections are also necessary. so it's not at all clear to me how the authors estimate the CALIOP detection likelihood without having some estimates of these quantities.

We changed the sentence to say, "It can be challenging to compare modelled aerosol concentrations to CALIPSO aerosol property information on a case-by-case basis. Doing so requires speculative assumptions about the lidar ratio and the extinction cross section of particles that are beyond the scope of this paper. Also, the amount of aerosol needed for CALIPSO to detect an aerosol layer is unknown and may vary over time and space, and the same can be said for the thickness of a CALIPSO layer required to be comparable with model aerosol concentrations. Therefore, our approach is instead to 1) assess how well large-scale Arctic CALIPSO combustion and dust aerosols are related to those in MERRA-2/FLEXPART, and 2) find and discuss where the largest discrepancies and similarities are between the products."

"Modelled aerosol concentrations are not directly comparable to CALIPSO direct backscatter and polarization observations and CALIPSO-inferred aerosol property or presence information. To derive a parameter such as aerosol concentration from CALIPSO, one would need to make speculative assumptions about the lidar ratio and the extinction cross section of particles, which is beyond the scope of this paper. Therefore, we focused on qualitative comparisons between dust and combustion aerosol distributions among MERRA-2, FLEXPART, and CALIPSO." can the matching problem be attacked from the other direction? that is, assuming the model provides a complete chemical and microphysical description of the aerosol loading with a fixed volume, can you not calculate an expected backscatter and extinction coefficient, which could then be used to develop a probability of detection by CALIPSO? (e.g., see Nowottnick et al., 2015; https://doi.org/10.5194/amt-8-3647-2015)

That's an interesting idea. Such an approach is outside the scope of this manuscript, but is definitely an idea worth considering for future work. However, it would still require assuming particle lidar ratios and extinction cross-sections (though in the model instead of in the CALIPSO analysis so at least the assumed particle properties are likely to be self consistent within the model).

" 'Polluted dust' was included in both the larger dust and pollution groupings." perhaps say why this is done and explain the consequences of having a single aerosol type with membership in both groups?

We added new panels in Figure 3 showing how the results would change if "polluted dust" is no longer included in the CALIPSO dust or combustion groups. We also added the following text to the methods in Section 2.2:

"Polluted dust made up 45% of the dust cases, and 39% of the combustion cases, and so there is significant overlap between the two groups. We discuss how removing the "polluted dust" component of the combustion aerosol group effects the results in section 3.3."

And we added the following text to the results in Section 3.3:

"Figure 3 also shows how the correlations change if "polluted dust" is excluded from the CALIPSO dust and combustion aerosol classification. Removing polluted dust did not have a large effect on relationships with combustion aerosols, but it reduced nighttime dust correlations for both MERRA-2 and FLEXPART at most altitudes, and substantially so for MERRA-2 near the surface. We conclude that polluted dust is an important contributor to total dust loads during polar night."

"Because this analysis focuses on aerosol concentration quantile values rather than on absolute concentrations, the aerosol concentration thresholds for determining a 'concentrated' aerosol layer are different between MERRA-2 and FLEXPART (see Table 2)." of course the problem with this is that CALIPSO is only sensitive to the absolute concentration. even if 100% of the aerosol within a sample volume is all the same type, if the absolute concentration remains too low, CALIPSO will not detect anything.

This text is related to the previous method for assessing model performance and is no longer included in the paper.

"Because CALIPSO has much finer vertical resolution than either MERRA-2 or FLEXPART, each CALIPSO profile was analysed within vertical bins comparable to either MERRA-2 or FLEXPART model vertical layers." I don't think you can do this in a genuine apple-to-apples way using the CALIPSO level 2 products. instead, the CALIPSO data averaging should be done at the level 1 stage, prior to launching the detections and classification schemes. if nothing else, the additional vertical averaging applied to the CALIPSO profiles will change the layer detection results (more averaging = better detection of fainter features), though perhaps at the risk of confounding some of the layer classification results (e.g., by vertical averaging over layers of different aerosol types; dust lying over marine is one prominent example).

The above text is now changed, along with the methods. It has been rephrased as follows:

"The analysis was done at model vertical resolution for FLEXPART (CALIPSO has much finer vertical resolution than either MERRA-2 or FLEXPART), and at 1 km vertical resolution for MERRA-2."

We also added the following text to the methods section:

"Furthermore, there is a risk that some layer classification results can be confounded by vertical averaging over layers of different aerosol types, such as dust lying over marine aerosol layers."

Although methodological improvements might be possible, we feel the current study demonstrates that the level 2 products are helpful for model validation. In these products, lidar data are averaged over relatively large areas, enabling standardized detection of dilute aerosol layers.

"The false positive (FP) rate is defined as:
$FPrate = \frac{n_{FP}}{n_c} * 100\%$    (3)

where $n_c$ is the number of CALIPSO observations with no aerosol of that subtype, and $n_{FP}$ is the subset of those data where modelled aerosol concentrations exceed the Table 2 threshold for that aerosol subtype."

- question #1: what counts as a "CALIPSO observation"? is this a 5-km x 30-m range bin, as might be inferred from the merged layer product? if so, what precautions were taken to avoid ambiguities and/or error that can be introduced by the (apparently) overlapping layers of different types that can be reported in the CALIPSO layer products? (e.g., see Thorsen et al., 2011; https://doi.org/10.1029/2011JD015970)

The aerosol layers in the CALIPSO profiles were obtained from the CALIPSO Lidar Level 2 5-km Merged Layer Data, V4-20 dataset, which is based on data with bins of 5-km x 30-m. We now include in the text that, "Sometimes multiple overlapping layers of different aerosol subtypes of interest were found (Thorsen et al., 2011). In those cases, the fraction of the model bin that was filled by a CALIPSO aerosol layer was determined by the sum of only the portions of each CALIPSO layer that did not overlap with other CALIPSO layers and that fell completely within the FLEXPART bin, divided by the entire height of the FLEXPART bin."

- question #2: are these "CALIPSO observations" required to be homogeneous? that is, the CALIPSO processing identifies and removes small scale clouds at single shot resolution in order to detect the aerosols in which these clouds are embedded? are these heterogeneous samples accepted or rejected in this analysis? based on line 185 ("cloud-free nighttime"), I assume they should all be rejected. are they?

Yes, as is now more clearly stated in the text, we are only looking at cloud-free profiles in this study.

- I suspect that using the aerosol profile product would likely have made the CALIPSO sample counting task much simpler and removed a large amount of the ambiguity in "sample purity" that can inadvertently occur when using the layer products.

Thank you for the suggestion. We will take that into account using the aerosol profile product instead of the aerosol layer product in potential future studies.

"Modelled aerosol layer presence was based on just a few aerosol species (i.e., BC and OC for combustion aerosol layers and super- and submicron dust for dust aerosol layers), but CALIPSO aerosol layer mixtures contain additional species that may vary relative to these constituents. Therefore, if for example, non-carbonaceous constituents in a plume observed by CALIPSO are present at high ratios relative to the modelled BC and OC species, that could lead to higher FN rates for combustion aerosols." so in this case CALIPSO detects an aerosol layer but classifies it as something other than a combustion aerosol? is that correct? i.e., the sample is a false negative because the model claims combustion aerosols are present but CALIOP says otherwise?

For clarity, we re-structured the information as follows:

"In addition to model error, limitations in the number of aerosol species being modelled vs. observed can also contribute to unexplained differences between MERRA-2/FLEXPART and CALIPSO. Real world CALIPSO aerosol layer mixtures contain additional species that may vary relative to the modeled constituents. For example, non-carbonaceous constituents in a combustion plume observed by CALIPSO can be present at high ratios relative to the modelled BC and OC and be mis-classified as carbonaceous."

"Conversely, if the ratios of non-carbonaceous constituents are low relative to the modelled species, that could lead to higher apparent FP rates for combustion aerosols. Similar trends would be expected for dust aerosols, if carbonaceous, biogenic, sulfate or maritime aerosols were elevated or reduced, respectively. However, we are not aware of any evidence to suggest that these uncertainties would lead to systematic biases in FN and FP rates, except near the surface over open ocean where marine particles are more common." ouch! to state the obvious, being unaware of evidence does not mean it does not exist. and it seems like there could be systematic biases if there were preferential aerosol mixing states; e.g., persistent injections of varying amounts of aerosol type X into an air mass regionally dominated by aerosol type Y. I'd think the models would be helpful in developing expectations about how often such scenarios might occur and whether the occurrence frequency was large enough to (potentially) introduce biases into the FN and FP rates.

We reworded as follows, "As another example, carbonaceous, biogenic, sulphate or maritime aerosols can be present in a dust plume. These constituents might lead to a bias in the MERRA-2/FLEXPART/CALIPSO comparisons at certain locations (e.g., over the open ocean surface downwind of a continental dust source)."

"CALIPSO aerosol presence in cloud-free profiles at the same altitude levels as the FLEXPART and MERRA-2 results is also shown in Figure 1. CALIPSO aerosol presence is not directly comparable to MERRA-2 and FLEXPART aerosol concentrations, which for example, can be impacted by high concentrations during infrequent events." I'd think this would be equally true for both measurements and (realistic!) models.

We have clarified (new text in bold): "CALIPSO aerosol presence in cloud-free profiles at the same altitude levels as the FLEXPART and MERRA-2 results is also shown in Figure 1. CALIPSO aerosol presence is not directly comparable to **average** MERRA-2 and FLEXPART aerosol concentrations, which for example, can be **skewed** by high concentrations during infrequent events."

Figure 1:
- having larger images would be a big help.
  Done.
- Regarding the CALIPSO plots: because polluted dust is included in both the "dust" and "combustion" classes, these are not disjoint plots
  True. However, there is overlap in dust and combustion layers in real aerosol layers, and so we think it is still appropriate to keep polluted dust in both (see new discussion in section 3.3). In addition, we have now re-done Figure 3 to showing how the results would vary when "polluted dust" is excluded from both plots. In short, removing polluted dust did not have a large effect on MERRA-2/FLEXPART relationships with CALIPSO combustion aerosols, but it reduced nighttime dust correlations for both MERRA-2 and FLEXPART at most altitudes, and substantially so for MERRA-2 near the surface, likely because pollution mixed in with dust is an important portion of the total dust load during polar night.
- Regarding the CALIPSO combustion plot: should one interpret this plot as saying that ~30% of the aerosols detected by CALIPSO are classified as "combustion aerosols"
  No, thanks for pointing out that this was confusing. We now explain in the revised Figure caption that one can interpret this figure to mean that CALIPSO-detected combustion aerosol layers occur ~30% of the time among all cloud-free CALIPSO observations over wintertime Siberia between 0.2 to 2 km asl.

"However, combustion aerosol layer distributions below 4 km are more sharply reduced over oceanic areas in CALIPSO than is predicted in the models. We suspect that this observation is caused by a CALIPSO aerosol subtyping artifact..."

while this shortcoming is well-known within the CALIPSO community, it may not have received sufficient attention in the literature. other than the Kanitz et al., 2014 work already cited by the authors, the only peer-reviewed papers I know of that acknowledge the issue are:

- Burton et al., 2013; https://doi.org/10.5194/amt-6-1397-2013
- Campbell et al., 2013; https://doi.org/10.1016/j.atmosres.2012.05.007
- Papagiannopoulos et al., 2016; https://doi.org/10.5194/acp-16-2341-2016 (only a very brief mention)
- Zeng et al., 2020; https://doi.org/10.3390/atmos12010010 (difference implicit in the maps shown in figure 9; certain types occur only over land (or ocean) and this leads to the discontinuities noted by the authors)
- (there may well be others too...)

Thanks for that useful information. We have added the new references into the paper:

"However, combustion aerosol layer distributions below 4 km are more sharply reduced over oceanic areas in CALIPSO than is predicted in the models. This observation is likely caused by a known CALIPSO aerosol subtyping artifact (Burton et al., 2013; Kanitz et al., 2014; Campbell et al., 2013; Papagiannopoulos et al., 2016; Zeng et al., 2021),…"

"others have found that polluted continental aerosols over the Arctic Ocean can be misattributed to clean marine conditions (Rogers et al., 2014; Di Biagio et al., 2018)." also see Oo and Holz 2011 (https://doi.org/10.1029/2010JD014894) for a potential solution to this kind of misclassification.

Great to see that reference, thank you! That is indeed useful to know going forward.

Section 3.2: The false positive and false negative calculations in section 2.2 report "the quality of the MERRA-2 and FLEXPART aerosol spatial distributions relative to those of CALIPSO". In this section I'd be interested to see similar assessment of CALIPSO aerosol type distributions relative to the expectation of diamond dust; that is, when diamond dust is likely in high concentrations, (a) how often does CALIPSO detect anything at all (i.e., cloud or aerosol) and (b) what fraction of those detections are classified as mineral dust.

Our study did not include cases where there were clouds because of the difficulty assessing aerosols above and below clouds. However, we now report in the new section in the supplement (and in the new Figure S2) the fraction of CALIPSO dust detections as a function of FLEXPART dust concentrations in conditions that do and do not favor diamond dust.

"However, conditions favorable for diamond dust formation from these pathways do occur frequently in some locations during the winter. At 925 mb, we estimate that wintertime diamond dust has the potential to confound the dust-CALIPSO comparisons up to 60% of the time (Fig. 2a), especially in low-lying areas of the Arctic Archipelago and the Siberian interior and coast (Fig. 2c)." to derive a useful estimate of CALIOP's misclassification frequency requires a prior estimate of detection capabilities. by design, the CALIOP retrieval scheme cannot misclassify aerosols that it does not detect. the likelihood of diamond dust occurrence is not sufficient; the authors must also establish the likelihood of detection by CALIOP

(which brings us back to my comments around lines 201-205). Maybe it would help to explicitly cast this discussion in terms of conditional probabilities???

To clarify, our main goal in this study was to evaluate FLEXPART and MERRA-2 using CALIPSO data and then use these products in concert with meteorological data to assess where certain cold-weather aerosol processes might be most important. We did not initially intend to estimate CALIOP's misclassification frequency. Our focus on diamond dust in section 3.2 was mainly to determine when and where diamond dust related issues were most likely to be a problem for the purpose of validating the model aerosol simulations using CALIPSO data. However, with the new methods, we did find that Kendall Tau and $R^2$ correlation metrics (see the new Figs. 3 and S3) are considerably worse in conditions that favor diamond dust formation.

Given that information and the reviewer's interest in this topic, we decided to provide some new related information in the supplement (also see the new Fig. S2):

"For the year 2008, we controlled for wintertime FLEXPART model dust concentrations (Fig. S2), and then compared the frequency of CALIPSO dust observations in conditions that were and were not favorable for the formation of diamond dust. We found that CALIPSO reported Arctic dust layers in wintertime air masses on average $61 \pm 11\%$ more often in conditions favorable for diamond dust formation ($RH_i > 100\%$) than in conditions not favorable for diamond dust. This finding was significant (Wilcoxon rank test, $p < 7e\text{-}13$) at all dust levels tested (Fig. S2), and similar differences were also seen in the later study years as well (data not shown). Given that conditions favorable for diamond dust occur up to ~60% of the time near the surface during the winter (Fig. 2a), this finding suggests that up to 37% of near-surface wintertime Arctic CALIPSO mineral dust observations overall could actually be diamond dust instead of mineral dust. This result assumes that the FLEXPART model is not biased between conditions with RHi values above and below 100%, that cloudy conditions would experience similar findings (since the current study focused on cloud-free conditions), and that these observed differences were attributable to diamond dust and not some other co-varying factor.
        However, conditions favoring diamond dust occur much less frequently than 60% of the time at higher altitudes, during the summer, and over most parts of the Arctic Ocean (Fig. 2). Therefore, we believe that diamond dust is unlikely to majorly impact Arctic CALIPSO dust data at most times and places."

[Regarding the "60% of the time" (above)]: I don't understand how to interpret this number. are the authors suggesting that diamond dust is present *in detectable quantities* ~60% of the time? if so, what's their metric for "detectable"? does this 60% number also account for the possibilities of mineral dust intrusions?

Thanks for pointing out that this was confusing. We have rephrased as follows:

"However, conditions favorable for diamond dust formation from these pathways do occur frequently in some locations during the winter, although this does not mean that diamond dust is always present and detectable by CALIPSO when conditions are favorable for its formation. At 925 mb, we estimate that conditions favorable to diamond dust formation occur up to 60% of the time (Fig. 2a), especially in low-lying areas of the Arctic Archipelago and the Siberian interior and coast (Fig. 2c)…. This means that wintertime diamond dust is most likely to confound the dust-CALIPSO comparisons at these times and places (as is discussed further in section 3.3 and Figure S2 of the Supplement)."

"We do not focus on daytime data because CALIPSO detects fewer aerosol layers during the day due to reduced sensitivity, which could cause FP rates to be biased high." I suspect there's still some useful info to be harvested from the daytime data; e.g., given that CALIPSO detects an aerosol (admittedly less frequently during day than at night). How often does the CALIPSO aerosol type classification agree with the models? the follow-on question would be whether the model-to-CALIPSO correspondence is essentially identical for both day and night or are there notable differences.

We rephrased the above sentence:

"Daytime data (shown for comparison in Figure 3, white backgrounds) appear to be slightly worse for MERRA-2 and FLEXPART dust at some altitudes but similar or maybe even slightly better for combustion aerosols. However, because CALIPSO detects fewer aerosol layers during the daytime, the remainder of our discussion is focused on nighttime data when the CALIPSO comparison data are of highest quality."

Figure 4: this is a very sweet set of plots. nice presentation (thumbs up for the discrete color bar) and high information content.

Thank you. We did unfortunately have to change the discrete color bar to help address a comment from Reviewer 1. Hopefully with information from the new methodological approach, information content in this figure is even higher.

"Although FLEXPART tended to somewhat overestimate dust concentrations at this site, MERRA-2 substantially underestimated observed mineral dust concentrations, particularly in the spring when local emissions are highest." eyeballing figure 5, I suspect that comparing RMS concentration differences between models and observations would show that MERRA2 has a smaller RMS difference than FLEXPART.

Thanks, this is a good point. We have rephrased as follows:

"Clearly both of these models leave something to be desired when it comes to matching the observations at this site. FLEXPART most of the time overestimates dust concentrations, whereas MERRA-2 underestimates dust throughout the year. Generally, the mean MERRA-2 dust concentration is closer than FLEXPART to the observed mean from July through March. However, FLEXPART dust variability is high enough to include the occasional extreme dust event observed episodically at this site, i.e., when dust concentrations exceed 100 µg dust m$^{-3}$ (Prospero et al., 2012). These events are most common in the spring when local emissions are highest. In contrast, a dust concentration of 100 µg m$^{-3}$ is many standard deviations outside the mean of the MERRA-2 values at this site. As such, it appears that MERRA-2 is less able to capture dust extremes than FLEXPART at this site, likely because it currently does not account for local dust sources."

"In contrast, FLEXPART, which does include local dust sources, is already able to predict the presence and absence of dust most of the time." but the FLEXPART predictions greatly overestimate measured dust concentrations in the fall and early winter

This has been noted in the text.

"Upon submission of this manuscript to ACPD, we will also submit the FLEXPART output for archiving at the NASA Earth Observing System Data and Information System (EOSDIS) data repository. If the

manuscript is accepted at ACP, a link to those output will be made available in the final manuscript." reviewers might have appreciated having the links in the discussion paper. I know for a fact that there are reviewers who routinely check those things as part of their review.

Thanks for the feedback. In the future we will ensure that there is at least a temporary link on a personal website that is available to the reviewers prior to publication. We are in the final stages of setting up a permanent link to the FLEXPART output at the NASA GES DISC permanent data archive (https://disc.gsfc.nasa.gov/). That link will be available very soon, and prior to typesetting, if the manuscript is accepted. In the meantime, the files can be found at the current website: https://folk.nilu.no/~nikolaos/ForLZ/

**References**

Di Pierro, M., Jaeglé, L., Eloranta, E. W., and Sharma, S.: Spatial and seasonal distribution of Arctic aerosols observed by the CALIOP satellite instrument (2006–2012), Atmos. Chem. Phys., 13, 7075–7095, https://doi.org/10.5194/acp-13-7075-2013, 2013.

Thorsen, T. J., Fu, Q., and Comstock, J.: Comparison of the CALIPSO satellite and ground-based observations of cirrus clouds at the ARM TWP sites, 116, https://doi.org/10.1029/2011JD015970, 2011.

---

## Referee Report (RR1)

In the revised manuscript, the authors have made substantial effort to improve the approach on assessing the model performance. I only have a few minor comments for the authors to consider.

I wonder how the Kendall Tau rank correlation metric is defined/calculated. It would be better to move lines 317-323 to line 210 to give the readers some ideas first. It is also not very clear to me how Z-scores are defined. $(X-\mu)/\sigma$?

The authors use "bin" in many places which refers to grouped/averaged range of different variables, like vertical levels, altitude, longitude/latitude. I would suggest using different word so that readers can know better what exact "bin" the authors are talking about.

Lines 208-210, it is not clear to me for the definition of the mean CALIPSO aerosol layer fraction. For one column during an altitude range, is it the number of vertical levels having the subtype of interest divided by the total number of vertical levels in this altitude range? It is the "overlapping with the vertical altitude bin of interest" making me confused.

Lines 219-222, I'm lost while reading it. Please try to rephrase it or cut it into several sentences.

Figure 3, I would suggest using the same x-axis range (-0.2 to 1.0?) for profiles both including "polluted dust" and excluding "polluted dust". It would be clearer to show that removing polluted dust has a small effect on relationships with combustion aerosols.

Figure 4, it would be better that some of the figure caption goes to the main text, which is more related to the discussion/interpretation of the results.

Lines 613-614, Thorsen et al. (2011) does not have the journal name.

---

## Author Response (AR2)

*Dear Hailong Wang, editor,*

*Thank you for accepting our paper for publication, pending technical corrections! Our corrections are listed below.*

*We also wanted to request the addition of a few new citations of relevant recent 2022 work that we have become aware of since the last submission. The citations and their textual context is provided below and does not change the conclusions or methodology of the paper. We believe adding these citations makes this a more cutting-edge contribution, and so we hope it will not be a problem to add them in. However, we also understand if that is not possible at this time. To that end, we have added the below text into the production file uploads for now, but will happily remove again it if it is deemed necessary. Thank you again for your consideration,*

*Sincerely,*
*-Lauren Zamora, on behalf of all the authors*

**Technical corrections**

In the revised manuscript, the authors have made substantial effort to improve the approach on assessing the model performance. I only have a few minor comments for the authors to consider.

*Thank you.*

I wonder how the Kendall Tau rank correlation metric is defined/calculated. It would be better to move lines 317-323 to line 210 to give the readers some ideas first. It is also not very clear to me how Z-scores are defined.

*The lines in question have been moved up, as suggested.*

*To clarify the software used in this analysis, which is relevant to how the Kendall-Tau and Z-scores were calculated, we added the following information to the beginning of section 2:*

"All data analysis was performed using the R language and environment for statistical computing version 4.2.0 (R Core Team, 2022)."

*We then added the following text to clarify how the Kendall Tau rank correlation is defined (new text in bold):*

"**The robust Kendall Tau rank correlation metric (τ) was chosen to assess correlations. It is calculated by examining all possible combinations of two data points in the data set and scoring them as either concordant (positive slope) or discordant (negative slope). τ is defined as:**

$$\tau = \frac{C-D}{C+D} \qquad (2)$$

**where C and D are the number of concordant and discordant pairs. In the case of ties, where a pair of data points is neither concordant nor discordant, a small correction is made ($\tau_b$, following Kendall (1945)).**

$\tau$ is a nonparametric, rank-based alternative to $R^2$ that is robust to outliers, makes no assumptions about the data distributions, and is thus a better metric of correlation for many types of data (Shevlyakov & Oja 2016). $\tau$ ranges from -1 for a perfect negative correlation to 0 for no correlation to +1 for a perfect positive correlation. In cases where the use of $R^2$ would be appropriate, the magnitude of $\tau$ is a good estimator for $R^2$, with a maximum theoretical asymptotic difference $< \pm 0.11$ (Shevlyakov and Oja 2016). For reference, Figure S3 shows corresponding $R^2$ values."

*As requested, we also added the following information to clarify how Z-scores are defined:*

"Next, we wanted to better understand where aerosol distributions between MERRA-2/FLEXPART and CALIPSO have the highest agreement. For this step, we assessed the difference in Z-scores between MERRA-2/FLEXPART concentrations and aerosol layer presence in CALIPSO. Z-scores in MERRA-2/FLEXPART for dust, BC, and OC are defined as the number of standard deviations from the mean respective dust, BC, or OC value across the study region at a given altitude level. Locations with high aerosol levels in MERRA-2/FLEXPART will have high positive Z-scores, and locations with low aerosols will have negative Z-scores. Similarly, Z-scores in CALIPSO are defined as the number of standard deviations from the mean dust or combustion aerosol layer fraction across the study region at a given altitude. Because Z-scores are unitless, they can be compared between MERRA-2/FLEXPART and CALIPSO, which has different units. This approach also enables relative comparisons between MERRA-2 and FLEXPART aerosol distribution patterns even if the concentrations are on different scales."

The authors use "bin" in many places which refers to grouped/averaged range of different variables, like vertical levels, altitude, longitude/latitude. I would suggest using different words so that readers can know better what exact "bin" the authors are talking about.

*Thank you, changed as suggested.*

Lines 208-210, it is not clear to me for the definition of the mean CALIPSO aerosol layer fraction. For one column during an altitude range, is it the number of vertical levels having the subtype of interest divided by the total number of vertical levels in this altitude range? It is the "overlapping with the vertical altitude bin of interest" making me confused.

*We clarified the text as follows:*

"To begin, we assessed the correlation between MERRA-2/FLEXPART aerosol concentrations and mean CALIPSO aerosol layer fraction. Correlations were calculated only on cloud-free days. CALIPSO aerosol layer fraction was defined as the fraction of the CALIPSO aerosol layer of the subtype of interest - dust or combustion – within each altitude bin for an individual CALIPSO observation. Then, as CALIPSO has much finer vertical resolution than either MERRA-2 or FLEXPART, CALIPSO dust layer fraction and MERRA-2/FLEXPART dust concentrations were

averaged within 20° longitude × 6° latitude bins and at model vertical resolution for FLEXPART, and at 1 km vertical resolution for MERRA-2. Similarly, CALIPSO combustion aerosol fraction was compared with MERRA-2/FLEXPART BC and OC concentrations. We excluded aerosol layers within 200 m of the surface in the lowest altitude bin and longitude-latitude-altitude bins with < 30 observations (collectively, < 1% of total data). Data were averaged either across the 8-year study period for one season (e.g., December through February) or for daytime/nighttime samples, as stated in the text.''

Lines 219-222, I'm lost while reading it. Please try to rephrase it or cut it into several sentences.

*Rephased as follows:*

"Sometimes CALIPSO had multiple overlapping layers of aerosol subtypes of interest, which is a result of CALIPSO's multiscale averaging approach (Thorsen et al., 2011). In those cases, we summed of only the portions of each CALIPSO layer that did not overlap with the other CALIPSO aerosol layer and that fell completely within the MERRA-2/FLEXPART bin. The sum of these portions was then divided by the entire height of the MERRA-2/FLEXPART bin to provide the fraction of the model bin filled by a CALIPSO aerosol layer."

Figure 3, I would suggest using the same x-axis range (-0.2 to 1.0?) for profiles both including "polluted dust" and excluding "polluted dust". It would be clearer to show that removing polluted dust has a small effect on relationships with combustion aerosols.

*We have changed the x-axis to be consistent in this figure, as suggested.*

Figure 4, it would be better that some of the figure caption goes to the main text, which is more related to the discussion/interpretation of the results.

*We have moved the following information to the text from the Figure 4 caption:*

"Because MERRA-2/FLEXPART and CALIPSO indicate aerosol presence in different units, they cannot be directly compared. The Z-score differences help locate where the overall patterns agree best between the two products. For example, if at a given location, MERRA-2 aerosols were three standard deviations above the mean and CALIPSO aerosol layer presence was only one standard deviation above its mean, that would result in a Z-score difference of two. The closer the Z-score difference is to zero, the more agreement there is between the products."

Lines 613-614, Thorsen et al. (2011) does not have the journal name.

*Fixed, thanks.*

**Proposed addition of new references and related text**

New text is shown in bold.

In section 2.1.2:

"There are several studies that have evaluated MERRA-2 Arctic aerosol distributions, mainly using MODIS aerosol optical depth and ground-based observations (e.g., Wu et al. (2020), Lee et al. (2020) and Sitnov et al. (2020)). MERRA-2 BC and OC aerosols tend to be a bit high compared to observed aerosol concentrations, although they tend to follow the qualitative trends (Vinogradova et al., 2020; Zhuravleva et al., 2020; **Xian et al., 2022**). One study found that dust optical depth and dust extinction were similar or a bit elevated compared to that of CALIPSO, but with large discrepancies in absolute concentrations (both over- and underpredicting concentrations) compared to two ground sites (Wu et al., 2020)."

In section 3.2:

"**It is important to note that the impacts on clouds of the dust and combustion aerosols of focus in this study may be much smaller than those of local marine emissions, particularly at times when the surface is not covered with snow or ice. For example, one recent satellite study found evidence that mixed phase cloud formation occurs above homogeneous freezing on average but at colder wintertime temperatures over both the Siberian and Canadian archipelago regions than over other times and Arctic locations (Carlsen and David, 2022), which the authors attribute to fewer marine ice nucleating particles being emitted there.**"

In the conclusions:
"**This study focused on dust and combustion aerosols. However, other recent studies indicate that marine aerosols may be particularly important for the Arctic ice nucleating particle budget (e.g., Carlsen and David (2022)). For example, Porter et al. (2022) found that biogenic particles may dominate the ice nucleating particle budget even over the Russian coast where we found that the dust aerosol impacts are likely to be highest. Thus, aerosol sources besides dust may enhance aerosol impacts on clouds over this region even further and could have a larger impact on clouds across the Arctic region than the long-range transported aerosols of focus in this study, particularly at times when the surface is not covered with snow or ice.**"

**References**

Carlsen, T. and David, R. O.: Spaceborne Evidence That Ice-Nucleating Particles Influence High-Latitude Cloud Phase, Geophysical Research Letters, 49, e2022GL098041, https://doi.org/10.1029/2022GL098041, 2022.

Porter, G. C. E., Adams, M. P., Brooks, I. M., Ickes, L., Karlsson, L., Leck, C., Salter, M. E., Schmale, J., Siegel, K., Sikora, S. N. F., Tarn, M. D., Vüllers, J., Wernli, H., Zieger, P., Zinke, J., and Murray, B. J.: Highly Active Ice-Nucleating Particles at the Summer North Pole, Journal of Geophysical Research: Atmospheres, 127, e2021JD036059, https://doi.org/10.1029/2021JD036059, 2022.

Xian, P., Zhang, J., O'Neill, N. T., Toth, T. D., Sorenson, B., Colarco, P. R., Kipling, Z., Hyer, E. J., Campbell, J. R., Reid, J. S., and Ranjbar, K.: Arctic spring and summertime aerosol optical depth baseline from long-term observations and model reanalyses – Part 1: Climatology and trend, Atmospheric Chemistry and Physics, 22, 9915–9947, https://doi.org/10.5194/acp-22-9915-2022, 2022.